# Canine Neonatal Assessment by Vitality Score, Amniotic Fluid, Urine, and Umbilical Cord Blood Analysis of Glucose, Lactate, and Cortisol: Possible Influence of Parturition Type?

**DOI:** 10.3390/ani12101247

**Published:** 2022-05-12

**Authors:** Tanja Plavec, Tanja Knific, Aleksandra Slapšak, Sara Raspor, Barbara Lukanc, Maja Zakošek Pipan

**Affiliations:** 1Veterinary Faculty, University of Ljubljana, Small Animal Clinic, Gerbičeva 60, 1000 Ljubljana, Slovenia; tanja.plavec@vf.uni-lj.si (T.P.); aleksandra.slapsak@gmail.com (A.S.); sara.raspor93@gmail.com (S.R.); Barbara.lukanc@vf.uni-lj.si (B.L.); 2Veterinary Faculty, Institute of Food Safety, Feed and Environment, University of Ljubljana, 1000 Ljubljana, Slovenia; tanja.knific@vf.uni-lj.si; 3Veterinary Faculty, University of Ljubljana, Clinic for Reproduction and Large Animals, Gerbičeva 60, 1000 Ljubljana, Slovenia

**Keywords:** neonate, Apgar score, lactate, glucose, cortisol, neonatal fluids

## Abstract

**Simple Summary:**

Parturition as a stressful event may influence puppies’ neonatal morbidity and mortality. The purpose of this study was to investigate the impact of parturition type on stress in newborn puppies, their weight gains, and survival in the first week postpartum. One hundred and twenty-three puppies were divided into three groups: vaginal parturition, emergency, and elective cesarean section. The Apgar score was assessed 5, 15, and 60 min postpartum, and samples of amniotic fluid, umbilical blood, and urine were collected for lactate, glucose, and cortisol concentration measurements. Although emergency cesarean section puppies had the highest cortisol concentration of all groups, their Apgar score at 5 min postpartum was comparable to the vaginal parturition group, which had the highest lactate levels. There were no significant differences between groups regarding relative growth rate. The type of parturition had no impact on puppies’ survival in our study, but supportive treatment was provided for non-vital puppies in stress. Non-invasive analysis of amniotic fluid and/or urine could help in the assessment of the neonatal vitality.

**Abstract:**

The objective of this study was to investigate the impact of parturition type on vitality in newborn puppies, their weight gains, and survival in the first week postpartum. One hundred and twenty-three puppies were divided in three groups: vaginal parturition (VP), emergency (EM-CS), and elective cesarean section (EL-CS). Apgar scores were assessed 5, 15, and 60 min postpartum. Lactate and glucose concentrations were measured in amniotic fluid and umbilical blood; cortisol concentrations were measured in amniotic fluid and puppy urine. Puppies’ weight gain was tracked daily for 7 days postpartum. Apgar score at 5 and 15 min was significantly better in the VP group. EL-CS puppies had significantly lower umbilical blood and amniotic fluid lactate concentrations compared to the VP group, which also had higher umbilical blood lactate concentration than EM-CS puppies. The cortisol concentration in the amniotic fluid and in urine differed significantly between the groups, with the highest concentration in the EM-CS, followed by the VP group. Glucose concentration in amniotic fluid was higher in the VP group than EM-CS group. The type of parturition had no impact on puppies’ weight gain or their survival at birth; however, supportive treatment was provided for non-vital puppies. Non-invasive analysis of puppies’ fluids could help in the assessment of the neonatal vitality.

## 1. Introduction

Parturition is always a challenge for the bitch and her puppies. The neonates are forced to adapt to their new life outside the womb in the first few hours after parturition [1]. Therefore, perinatal disorders and losses in dogs are a common and often unavoidable problem; they can occur in utero, during labor, immediately after, or within the first two to three weeks of life, but most puppy losses occur during whelping and the first 24 h [2]. Many factors may contribute to higher neonatal mortality in dogs [3], ranging from 5% to 35%, depending on complications before, during, and/or after parturition [4,5,6,7,8]. Adequate reproductive management throughout the bitch’s reproductive cycle and good veterinary care at the onset of parturition, with intensive neonatal care and monitoring during the first days of life, can significantly reduce neonatal losses [9,10,11]. At the time of weaning, 91% of 1342 newborn puppies were alive in a study by Münnich and Kuechenmeister, reflecting the aforementioned reproductive management [8].

In human medicine, neonatal assessment is performed using the Apgar score, which immediately assesses the infant’s status at birth and targets the need for neonatal resuscitation [12]. In 2009, the Apgar score was modified to adapt it to the needs of canine neonates [13]. Observation of the neonate and measurement of vital signs via the modified Apgar score in conjunction with reflex assessment in the first hour of life has been shown to be satisfactory for assessing neonatal status [12] and short-term survival prognosis [13]. However, in practice, clinical evaluation of neonatal puppies has been largely limited to a subjective examination. Basic diagnostics to objectively evaluate neonatal puppies are readily available to practitioners, but without established parameters, it is difficult to analyze them and determine what is normal versus abnormal and what might indicate future vitality [14]. Further, the type of parturition can also have a major impact on the vitality and survival of puppies after parturition [5,11], with puppies delivered by cesarean section (CS) having better survival than naturally born puppies [5].

In humans, the Apgar score is combined with umbilical blood gas analysis, which provides important information about the condition of the neonate [15]. There are limited data on the blood parameters in neonatal dogs [16,17], but as in humans, measurement of specific biomarkers from umbilical blood or amniotic fluid, e.g., lactate, cortisol, and glucose, could help to discriminate between healthy neonates and those requiring veterinary assistance [10,18]. Since these parameters are good indicators of non-vital neonates in humans, they were chosen in our study to see if they could also be used as indicators of neonatal vitality in puppies.

Clinically, it has been demonstrated that very high cortisol concentrations in puppies in amniotic fluid are a reliable indicator for identifying those neonates that require intensive care in the first 24 h of life [10]. In a recent study, all groups of puppies had cortisol concentrations above the basal concentration at birth. Significantly higher cortisol levels were noticed in fetal dystocia puppies compared to eutocia, maternal dystocia, and CS puppies. The difference between fetal dystocia and CS puppies persisted at 60 min after birth [19].

Increased lactate concentrations in the blood indicate the presence of anaerobic cellular metabolism, which is a sign of tissue hypoperfusion and hypoxia. Lactate concentrations increase before any abnormalities in heart rate, blood pressure, or urine output occur, making blood lactate measurement a superior method for early detection of hypoperfusion [20]. In dogs, there is a study that examined the concentration of lactate in the umbilical vein, which was found to be useful in predicting neonatal mortality within 48 h of birth. The threshold of 5 mmol/L of umbilical vein lactate concentration allowed differentiation between healthy and non-vital puppies. Higher lactate values were associated with non-vital puppies, whereas lower values characterized vital puppies [17]. Mean blood lactate levels of 10.0 mmol/L ± 4.9 standard deviations (SD) were found in non-surviving puppies [21]. However, this was not confirmed in two other studies where lactate levels did not vary between vital and disturbed puppies [22,23].

Transient hypoglycemia is common in neonates. Insufficient glycogen storage at parturition is a frequent occurrence in puppies with low birthweight and those suffering from respiratory failure or severe respiratory distress during parturition; hence, the measurement of glucose concentration may aid as a prognostic factor [24]. In one study on puppies evaluating glucose concentrations from ear blood, it was found that low blood glucose concentration was found in puppies with higher risk of neonatal mortality [25]. However, Lucio et al. (2021) found that fetal dystocia and CS are hyperglycemic obstetrical conditions for neonatal puppies [19]. There is also a study dealing with amniotic glucose concentration in newborn puppies where it was found that non-surviving puppies had lower median amniotic fluid glucose concentrations compared with neonates that survived. However, two of six amniotic fluid glucose concentrations were below the minimum detectable concentration, so this result should be taken in consideration with caution [23].

The limited data, the different ti”es o’ puppy evaluation or collection of samples, and the different vessels used for blood collection make it very difficult to compare the results mentioned above. A thorough evaluation of non-invasive parameters and their correlation with neonatal vitality are still lacking in veterinary neonatology. Therefore, the aim of this study was to assess lactate and glucose concentrations in newborn umbilical blood and fetal amniotic fluid, cortisol concentrations in fetal amniotic fluid, and urine of newborn puppies, and to evaluate the association between these biomarkers with newborn vitality score, parturition type (vaginal parturition (VP) versus elective CS (EL-CS) versus emergency CS (EM-CS)), and puppy survival within the first seven days of life. We also aimed to investigate the association between parturition type and puppy survival and growth. The vitality of the newborn puppies was assessed using the modified Apgar scoring for dogs.

## 2. Materials and Methods

### 2.1. Animals

The study was approved by the Animal Welfare Commission of the Veterinary Faculty. The certificate is included as a Non-published Material. All animals were client-owned: the owners volunteered their dogs for the purpose of this research and signed a consent form allowing for the collection of samples from the bitch and the newborns. Samples were collected from March 2017 to May 2018. During this time, 123 puppies born to 20 healthy bitches of 16 different breeds (two Boston terrier bitches whelped twice) were included in our research. The mean age of the bitches was 41.6 months ± 19.2 SD (range: from 23 to 86 months). All bitches were fed only FDA-approved food for pregnant dogs. Bitches for VP and EM-CS were not fasted before the parturition. The bitches for EL-CS were also allowed to eat a small amount of soft canned food 3 h before the planned surgery. The number of puppies ranged from two to 11 per litter (median 5.5). The mean litter size was 5.59 puppies ± 2.38 SD. The types of parturition were divided into vaginal parturition (VP) and caesarian section (CS). CS was further divided into emergency CS (EM-CS) and elective CS (EL-CS). Regarding the type of parturition, there were 9 VPs, 8 EL-CSs, and 5 EM-CSs. One bitch gave birth to two alive puppies and one dead puppy via VP and was then presented for the EM-CS (she and the following 4 puppies were accounted to the EM-CS group) (Table 1).

For survival data, each newborn was categorized as stillborn or born alive.

### 2.2. Cesarean Section

This group consisted of EM-CS and EL-CS. If the medical management of dystocia had failed or was inadvisable, the EM-CS was performed according to indications as previously described [26,27]. The reason for the EM-CS was the primary uterine inertia in one dam, secondary uterine inertia in two, and obstruction in the other two. In one case, the puppy was too big and in the other it was in an incorrect posture. All dams were in good general condition. However, at least one puppy was under stress in all cases, as indicated by its heart rate, measured with ultrasound. The bitch with primary uterine inertia was given oxytocin twice (0.25 IU and 30 min lates 0.5 IU), prior to coming to the clinic for EM-CS.

EL-CS was planned based on progesterone measurements during heat to determine LH peak and ovulation, coupled with ultrasound (US) examinations of the ovaries and uterus. The day of surgical parturition in EL-CS cases was determined using the average 63-day gestation from ovulation guidelines, coupled with a decrease in the dams’ rectal temperature and the information collected from the serial blood progesterone concentration monitoring and by fetal ultrasonographic measurements of both the inner chorionic cavity and the biparietal diameter, as reported by Meloni [28]. None of the puppies showed signs of immaturity at birth.

CS was always performed using the same anesthetic protocol. The preoperative health status of the bitch was determined via a clinical examination and complete blood count (ADVIA^®^ 120, Siemens, München, Germany) with blood collected from a peripheral vein. The cephalic vein was used for intravenous cannulation. During the preoxygenation with 100% oxygen for 10 min, the abdomen was shaved and aseptically prepared in a routine manner. Propofol at 4–7 mg/kg of body weight (BW) was given intravenously (Propoven 10 mg/mL, Fresenius Kabi Austria GmbH, Graz, Austria) for induction to anesthesia, followed by orotracheal intubation. General anesthesia was maintained with sevoflurane (Sevorane, AbbVie, Campoverde di Aprilia, Italy) at a concentration of 1.5–3%. Methadone (Synthadon 10 mg/mL, Produlab Pharma B.V., Raamsdonksveer, The Netherlands) at 0.2 mg/kg BW was administered subcutaneously after the skin incision. A balanced isotonic crystalloid solution (Hartmann solution, B. Braun, Melsungen, Germany) at a rate of 10 mL/kg/h intravenously was started 30 min before general anesthesia. CS was performed with a caudal midline abdominal incision, followed by a ventral incision of the uterine body. The most caudal fetus and its fetal membranes were removed first, followed by fetuses and fetal membranes from the left and right uterine horn alternately.

A simple continuous suture pattern using 3/0 or 4/0 Glycomer 631 (Biosyn, Covidien, Dublin, Ireland) was used to close the uterus. To promote uterine contractions and cleaning, oxytocin (Oxytocin Veyx, Veyx-Pharma Gmbh, Schwarzenborn, Germany) at 0.25–1 IU was injected into the uterus at the end of the celiotomy. The abdominal cavity was thoroughly inspected and rinsed with 0.9% sodium chloride (NaCl, B Braun, Melsungen, Germany) and warmed to body temperature before the abdominal wall was sutured routinely in three layers. Lidocaine 10 mg/mL (Lidocaine HCl, University Medical Centre, Ljubljana, Slovenia) at a dosage of 1 mg/kg was administered infiltrative at the incision line.

Four hours after the first dose, methadone (0.2 mg/kg) was administered subcutaneously. When the mothers were awake and demonstrated normal maternal behavior, the bitch and her puppies were discharged from the clinic. Postoperative analgesia was provided with tramadol chloride (Tramal, Stada, Bad Vilbel, Germany) at 3 mg/kg orally every 12 h, as needed, for a maximum of three days postoperatively. When the bitch did not have colostrum at the time of parturition, the puppies were given a drop of 40% glucose under the tongue. If mammary secretions were still not available, newborns were fed with a commercial milk replacer formula (Puppy Protech Colostrum + Milk, Royal Canin, Ljubljana, Slovenia) at suggested dosages every two hours. This was needed only for one litter of EL-CS puppies (for the first day) and for one other puppy that died one day postoperatively.

#### Puppy Resuscitation

The basic steps for puppies who did not breathe spontaneously and did not vocalize followed the resuscitation guidelines of ABC. First, the airway was cleared by removing the fetal membranes from the face, followed by gentle suctioning with a bulb syringe. The puppies were then gently but briskly dried with a warm towel to promote respiration and prevent chilling. If the neonates were still not breathing, a Jen Chung vacupuncture point in the nasal philtrum with a 26-gauge needle was used. For neonates who were not breathing spontaneously, a constant flow of oxygen was administered through the oxygen mask. Ventilation or endotracheal intubation and cardiac stimulation were not required. The only drug used was naloxone (one drop under the tongue) when the puppies did not respond to environmental stimuli. Puppies were kept warm during resuscitation and in the immediate postpartum period. Water bottles heated to 38 °C were used for this purpose.

### 2.3. Vaginal Parturition

Vaginal deliveries took place in the home environment of the owner.

### 2.4. Evaluation of the Puppy, Apgar Score, and Neonatal Reflexes

Each newborn puppy was weighed, assessed using the modified Apgar scoring system [14,17,29], and examined for the presence of congenital malformation (congenital oronasal fistula, cleft palate, atresia ani). When needed, resuscitation was performed by ABC route: rubbing to stimulate breathing, cleaning mucous from the upper airways, oxygen supplementation, heating, glucose supplementation, and, on one occasion, external heart massage. Each puppy was marked immediately after parturition with a colored collar to enable the identification of the puppies and correctly mark the collected samples.

The vitality of the puppies was assessed by observing the color of the mucous membranes, auscultation of the heart to measure the heart rate, and evaluation of the frequency and quality of respiration. The neonatal irritability reflex was assessed with a firm, almost painful stimulus on the back. Active movements of the puppies were observed in the dorsal position to assess muscle tone. Each parameter was scored as 0, 1, or 2 points (Table 2). The Apgar score was interpreted depending on the number of points collected, severe distress from 0 to 3, moderate distress from 4 to 6, and no distress from 7 to 10 points. Apgar score was evaluated 5 (Apgar 5), 15 (Apgar 15), and 60 (Apgar 60) min after parturition.

Neonatal reflexes (suckling, rooting, and righting reflex) were also assessed at 5, 15, and 60 min after parturition. The suckling reflex was assessed by inserting a clean, warm tip of the smallest finger into the mouth of the puppies. The righting reflex was assessed by turning the puppies onto their backs and observing how quickly they turned back to the sternal position. The rooting reflex was scored by holding a loosely clenched fist in front of the puppy and observing whether the puppy responded by pressing its snout into the hand. Each reflex was scored from 0 to 2, with 0 representing no reflex and 2 representing a strong reflex. Then the total sum was calculated. The interpretation of the results was as follows: poorly responsive from 0 to 2, moderately responsive from 3 to 4, and adequately responsive from 5 to 6 points [12,14].

### 2.5. Amniotic Fluid, Umbilical Blood, and Urine Samples

Samples of amniotic fluid were taken immediately after birth from all puppies, but only samples of alive puppies were processed further. The puppy was put in an upward position, the amniotic sac was freed from the puppy’s head, and the amniotic fluid samples were collected from the underlying amniotic sac in an aseptic manner using a sterile needle and syringe. Then the amniotic sac was opened completely, and the amniotic fluid was decanted from a syringe in a urine collection tube. The umbilicus was clamped on two sides and a small 25-gauge needle was used to try to obtain umbilical blood within 2 min after birth. At least 25 µL of umbilical blood was taken and was processed immediately. Amniocentesis, sampling of the umbilical blood, and neonatal resuscitation were performed by different individuals.

Amniotic and umbilical blood samples of newborns delivered by spontaneous whelping were collected the same way as in the CS group, if they were born with an intact amniotic sac and umbilical cord. Urine samples were collected within an hour after parturition after manual stimulation of the vulva/prepuce of the neonate with the moistened swab. The first two drops of urine were discarded.

Amniotic fluid analysis included measurement of specific gravity with refractometer (Veterinary refractometer, Eickemeyer, Tuttlingen, Germany), glucose (Wellion Gluco Calea, Med Trust Handelsges.m.b.H., Marz, Austria), lactate (Accutrend Plus, Roche, Basel, Switzerland), and cortisol concentration (MiniVidas analyzer, BioMerieux S.A., Lyon, France) in accordance with the manufacturer’s instructions. For cortisol measurement, the amniotic fluid sample was centrifuged at 2000× *g* for 10 min and frozen within an hour at −80 °C until analyzed less than 6 months after collection.

Umbilical blood analysis included glucose (measured with Wellion Gluco Callea), and lactate concentration (measured with Accutrend Plus), and the samples were processed within the first 5 min.

After the first suckling, samples of newborn urine were collected. Urine was frozen at −80 °C within an hour and stored for no longer than 6 months. Cortisol concentrations were then measured using the MiniVidas analyzer.

At the beginning of the study, we performed a partial validation of amniotic glucose and amniotic lactate assay by determining the accuracy and repeatability. The accuracy of the methods for lactate measurements was determined by analysis of a control sample with known low- and high-lactate concentrations, 3.6 and 9.0 mmol/L, respectively. The measured values were 3.32 mmol/L ± 0.12 SD and 8.49 mmol/L ± 0.21 SD; the inter-assay coefficients of variation (CV) for the control solution were 3.42% and 2.51%, respectively. By performing 6 measurements of the single amniotic fluid sample, the intra-assay CV for lactate was 3.21% and 2.15% for the two samples with 4.4 (low) and 23.3 mmol/L (high) lactate concentrations, respectively. The procedure of measuring intra-assay CV was repeated several times during the study: If there was enough amniotic fluid, the coefficient of variation stayed in the mentioned frame.

By performing 6 measurements of the single sample, the intra-assay CV for glucose in amniotic fluid was 3.73%. Since there were no puppies with high amniotic glucose concentrations, only one CV was calculated.

At the beginning of the study also a partial validation of amniotic fluid and urine cortisol, determining the accuracy and repeatability was performed. The accuracy of the methods for cortisol measurements was determined by analysis of a control sample with known low- and high-cortisol concentrations, 0.725 and 22.112 ng/mL, respectively. The measured values were 0.729 ng/mL ± 0.011 SD and 22.009 ng/mL ± 1.003 SD; inter-assay CVs for the control solution were 1.52% and 2.23%, respectively. By performing 6 measurements of the single sample, the intra-assay CV for cortisol in amniotic fluid was 2.74% and 1.35% for two samples with 8.79 (low) and 16.37 ng/mL (high) cortisol concentrations, respectively. Six measurements of the single urine sample were performed and the intra-assay CV of variation for cortisol in urine was 2.08% and 1.23% for two samples with 11.03 (low) and 18.49 ng/mL (high) cortisol concentrations, respectively.

### 2.6. Growth Rate of Newborn Puppies

Immediately after birth, the puppies were weighed on a steady scale with an accuracy of +/−1 g. After the first day, the puppies were weighed by the owners on the same scale. The growth rate of the puppies was followed until they were eight weeks old. The bodyweight of the puppies was measured each morning in the first week. Later, they were weighted once weekly in a 7-day interval. The day of the parturition was determined as Day 0.

The relative growth rate was calculated as the relative change in the bodyweight compared to the puppy’s initial birthweight.

### 2.7. Statistical Analysis

Data were collected and edited in Microsoft Excel 365. The statistical analyses were performed using R statistical software, version 4.1.2 [30], and *p* < 0.05 was considered significant.

The data are presented with basic descriptive statistics; e.g., median, quartiles, mean ± standard error of the mean (SEM). The normal distribution of the data was tested by the Shapiro-Wilk test. Differences between blood parameters, amniotic fluid, urine, puppies’ vitality estimates, and relative growth rates according to the type of parturition and vitality were tested with a non-parametric Kruskal–Wallis rank-sum test and Wilcoxon rank-sum test, respectively. Due to multiple comparisons, we adjusted the *p*-values with a Benjamini–Hochberg correction. Correlations were given using the Spearman correlation coefficient, and statistical significance was determined according to the adjusted *p*-value according to the Holm method. The interpretation of Spearman’s correlation coefficient is as follows: 0.00–0.19—very low/weak; 0.20–0.39—low/weak; 0.40–0.59—medium/moderate; 0.60–0.70—high/strong; 0.80–1.0—very high/strong correlation.

## 3. Results

### 3.1. Basic Information

The data of different breeds, included in our research, and the number of puppies per breed are presented in Table 1. The number and proportion (%) of puppies according to the type of parturition, sex, and survival of newborns are presented in Table 3. The birthweight data are missing for two puppies that were born at home via VP before the bitch was presented for EM-CS; additional six puppies were lost to follow-up after day 0.

### 3.2. Survival, Apgar Score, and Neonatal Reflexes

The stillborn puppies (*n* = 9) were excluded from the statistical data analysis. Seven puppies (5.7% of all puppies or 6.1% of born alive) from four bitches died after the discharge from the clinic: They all died within the first week of life (only one died in the first 48 h) and all but one were in severe distress 5 min after parturition. At 15 and 60 min postoperatively, two and five of them, respectively, were in no distress. In one puppy, Apgar 5 was 1, Apgar 15, and Apgar 60 was 3; the puppy was born via VP and died one day postpartum. One puppy showed no distress at 5, 15, and 60 min postoperatively and died on the fourth day postoperatively of an unknown cause. One of the puppies that died on the third day postpartum was diagnosed with neonatal sepsis due to infection with *Staphylococcus pseudintermedius*. Most likely, the puppy was infected through breast milk, although the bitch did not show any clinical signs. Namely, a pure culture of the same bacterium was isolated from its milk [31]. Three Pembroke Welsh Corgi puppies were diagnosed with interstitial pneumonia postmortem, and no infectious agent was found. The others were not examined by autopsy and the cause of death remains unknown.

The results of the Apgar score 5, 15, and 60 min after parturition are presented in Table 4. The data are missing for two puppies that were born at home via VP before the bitch was presented for EM-CS. Data are also missing for 4 puppies at 5 and 15 min and for 3 puppies also at 60 min. These puppies were born with VP before we arrived at the parturition site.

The median Apgar score was 8 for Apgar 5, 9 for Apgar 15, and 10 for Apgar 60.

The most vital puppies were born with VP, followed by puppies born with EM-CS and EL-CS. The Apgar score 5 min after parturition for EL-CS was significantly lower when compared to EM-CS and VP (*p* < 0.0001) (Figure 1A).

Puppies born with VP still had the highest Apgar scores 15 min after parturition compared with both other groups (*p* < 0.0001), but there was no statistical difference when EL-CS and EM-CS were compared (Figure 1B). There were no statistical differences in Apgar scores between groups 60 min after parturition (*p* = 0.2961). Median Apgar scores of 10 were observed in all three groups.

In the EL-CS group, 26 puppies (72%) were in severe distress 5 min after parturition, whereas 54 (95%) and 10 (67%) of puppies born with either VP or EM-CS showed no distress. Differences in the distribution of Apgar scores were also observed 15 min after parturition, with the distribution of the puppies born with EM-CS and VP similar to 5 min after parturition. However, most puppies born with EL-CS were not in distress and the proportion of severely distressed puppies born with EL CS decreased compared with Apgar scores 5 min after parturition. As a result, the proportion of puppies being moderately distressed 15 min after parturition increased compared with the Apgar score 5 min after parturition.

Puppies in severe distress 5 and 15 min postpartum had statistically significantly worse survival in the first week postpartum, *p* = 0.0113 and *p* = 0.0231, respectively.

Table 5 presents the number of puppies with a certain reflex score according to the type of parturition 5, 15, and 60 min after birth. Stillborn puppies were excluded from the statistical processing of reflex assessment. Data for 6 puppies at 5 and 15 min and data for 5 puppies at 60 min as for Apgar score are missing.

### 3.3. Amniotic Fluid, Umbilical Blood, and Urine Samples

The mean specific gravity of the amniotic fluid was 1.012 kg/L ± 0.004. The mean value for EL-CS was 1.011 kg/L ± 0.005 for EM-CS 1.015 kg/L ± 0.005 and for VP 1.012 ± 0.004. There were no significant differences between the values of the specific gravity of amniotic fluid between the different parturition types.

The mean cortisol concentration of amniotic fluid (*n* = 87) was 8.14 ± 0.48 ng/mL, and the median was 8.70 ng/mL. The lowest concentrations of cortisol were observed in the EL-CS group (3.83 ± 0.44 ng/mL), followed by the group of puppies born via VP (9.24 ± 0.36 ng/mL), and the EM-CS group (14.01 ± 1.10 ng/mL). The difference in cortisol concentration between all three types of parturition was significant (*p* < 0.0001) (Figure 2A).

The mean cortisol concentration in urine (*n* = 87) was 9.64 ± 0.54 ng/mL, and the median was 9.30 ng/mL.

The lowest mean urinary cortisol concentration was observed in puppies born with EL-CS (4.33 ± 0.41 ng/mL), followed by the group of puppies born with VP (10.62 ± 0.37 ng/mL) and EM-CS (16.70 ± 1.05 ng/mL). The difference in urinary cortisol concentration between all three types of parturition was significant (*p* < 0.0001) (Figure 2B).

The mean glucose concentration in the amniotic fluid (*n* = 55) was 3.49 ± 0.19 mmol/L, and the median was 3.70 mmol/L. The lowest mean glucose concentration in the amniotic fluid was observed in puppies born with EM-CS (*n* = 11; 2.73 ± 0.72 mmol/L), followed by puppies born with EL-CS (*n* = 6; 2.80 ± 0.58 mmol/L), and those born with VP (*n* = 38; 3.81 ± 0.15 mmol/L). The difference in the amniotic fluid glucose concentrations between parturition types was not significant (*p* = 0.0666). No statistically significant differences in blood or amnion glucose concentrations (*p* = 0.1775 and *p* = 0.2090, respectively) were found between dead and live pups.

The mean umbilical blood glucose concentration (*n* = 59) was 3.68 ± 0.17 mmol/L, and the median was 4 mmol/L. Puppies born with EL-CS (3.23 ± 0.38 mmol/L) had the lowest mean glucose concentration, followed by puppies born with EM-CS (3.29 ± 0.72 mmol/L) and those born with VP (3.80 ± 0.15 mmol/L). There were no significant differences between the concentrations of umbilical blood glucose between the different types of parturition (*p* = 0.1652).

The mean lactate concentration in the amniotic fluid (*n* = 68) was 11.32 ± 0.54 mmol/L, median 10.63 mmol/L. Puppies born with EL-CS (9.11 ± 0.87 mmol/L) had the lowest mean lactate concentration, followed by puppies born with EM-CS (10.25 ± 1.82 mmol/L) and VP (12.24 ± 0.56 mmol/L). The difference in the amniotic fluid lactate concentration between the EL-CS and VP was significant (*p* = 0.0267) (Figure 3A).

The mean lactate concentration in umbilical blood (*n* = 66) was 8.39 ± 0.55 mmol/L, median 8.15 mmol/L. The lowest mean umbilical blood lactate concentration was observed in puppies born with EL-CS (3.90 ± 0.58 mmol/L), followed by puppies born with EM-CS (4.23 ± 0.41 mmol/L) and VP (10.41 ± 0.57 mmol/L). The difference in umbilical blood lactate concentration between EM-CS and VP as well as between EL-CS and VP was significant (*p* < 0.0001) (Figure 3B).

Puppies born at the end of the parturition did not have significantly different lactate concentrations in either umbilical blood or in amniotic fluid (*p* = 0.1775 and *p* = 0.2751, respectively).

### 3.4. Growth Rate of Newborn Puppies

The bodyweight of puppies at birth and on day seven by parturition type and separately for female and male puppies are shown in Table 3. In Figure 4, we present the relative growth rate from day one to seven by the type of parturition. The mean relative growth rate on day 1 was −1.95 ± 0.44%, on day 2: 3.52 ± 0.54%, on day 3: 12.55 ± 0.59%, on day 4: 23.76 ± 0.48%, on day 5: 34.49 ± 0.49%, on day 6: 48.26 ± 0.44%, and on day 7: 63.09 ± 0.39%.

Puppies born by EL-CS had the lowest relative growth rate, and puppies born by VP had the highest relative growth rate; however, there were no statistically significant differences between the three types of parturition on any given day (*p*: 0.0878–0.9515) (Figure 4).

The growth rate of the puppies was followed until they were eight weeks old, but a significant difference in relative growth rate between puppies was not observed.

### 3.5. Correlations

The parameters measured in the blood and in the amniotic fluid were compared with each other and their correlations were determined (Table 6).

Highly statistically significant (*p* < 0.0001) and very strong correlations were found between the cortisol concentrations in urine and in amniotic fluid and also between the lactate concentrations in amniotic fluid and in umbilical blood. There was a strong correlation between Apgar 5 and Apgar 15, which was also highly statistically significant (*p* < 0.0001). However, Apgar 60 did not correlate with any of the observed parameters. The umbilical blood glucose moderately correlated with glucose concentrations in the amniotic fluid (*p* = 0.0170), and with Apgar score 5 min (*p* = 0.0093) and 15 min (*p* = 0.0036) after parturition. Glucose in the amniotic fluid, however, moderately correlated with Apgar score 5 min after parturition (*p* = 0.0423). Lactate in the umbilical blood moderately correlated with cortisol in urine (*p* = 0.0045). The relative growth rate in puppies on day 1 moderately correlated with lactate in the amniotic fluid (*p* = 0.0144) and lactate in the umbilical blood (*p* = 0.0035). However, the relative growth rate from the second and later days did not correlate with any of the other variables.

## 4. Discussion

The objective of this study was to explore neonatal puppy vitality by investigating common clinical parameters with Apgar score and determining glucose, lactate, and cortisol parameters in fetal and neonatal (urine) fluids while comparing different parturition types. The clinical parameters assessed in this study were selected because they are widely used in veterinary medicine for the subjective assessment of puppies to determine their vitality. These parameters, together with the rapid assessment of the above-mentioned laboratory parameters, could serve as a good predictor for identifying puppies that need special care during the first hours or days after birth in order to improve their survival rates.

In our study, 9 (7.3%) of the puppies were stillborn and 7 (5.7%) died later. This is in accordance with other recent studies [5,6,9,13,17,32] reporting mortality rates of as high as 25% in the clinical setting [5,6]. Better survival rates are reported in controlled pregnancies and parturitions, suggesting that survival can be improved by appropriate reproductive management and support during pregnancy and parturition, and by timely surgical intervention when needed [9,13,17,32]. However, the most important question is how to identify the newborns at risk and help them in the first days of life. Birthweight [6,32,33,34] and Apgar score [32] are recognized as prognostic factors for early postpartum survival. On the contrary, as reported in [34], the Apgar score seems to be more accurate. Of the seven puppies that died in the first week postpartum in the present study, six were in severe distress 5 min after birth. A statistically lower Apgar score measured at 5 and 15 min after birth was significantly associated with higher mortality. Similarly, an Apgar score between 0–3 measured 5 min after birth was associated with a higher mortality rate than scores between 4–6 and higher in previous studies [13,29,32,35]. In our study, the condition of the puppies improved over the next 10 and 55 min, respectively, indicating the need for an early Apgar score to identify at-risk neonates, as a later measurement may mask the initial stress level, as in our study 99% of puppies showed no distress 60 min after birth. The Apgar score did not correlate with the relative growth rate in our study.

Thirty-one puppies (28.7%) in this study population showed severe distress 5 min after birth, which is significantly more than in some other studies, reporting between 85.3% and 94% normally viable puppies 5 min after birth [10,13], but similar or less than 17% at 5 min [36] or 45.3% of puppies showing severe distress 10 min after birth [17]. In both studies [17,36], puppies born with CS predominantly contributed to Apgar scores 0–3 at 5–10 min after birth, demonstrating the differences in Apgar scores between the different types of parturition. Sixteen of thirty-seven puppies (43.2%) born with CS showed severe distress 5 min after birth [36], but Groppetti et al. (2010) reported as many as 100% and 92% of puppies showing severe distress 10 min postpartum in the EM-CS and EL-CS groups, respectively [17]. Our study confirms significant differences between the groups 5 and 15 min after birth where 72% of the EL-CS puppies, but only 13.3% of the EM-CS and 5.3% of the VP puppies, showed severe distress 5 min postpartum. The results of puppies’ responsiveness were very similar to the results of the Apgar score. While anesthesia could be a possible factor for the lower Apgar scores and poor responsiveness in all CS puppies [35,37], the very high proportion of the EL-CS puppies in severe distress in our study compared to two previous studies [10,32] requires further investigation. The reason for the lower Apgar scores could be influenced by the determination of the parturition date, although we used several parameters to determine the best timing for EL-CS. However, their accuracy depends on several factors such as breed, gestational age, and litter size. Therefore, it is challenging to choose the most appropriate parameter for all breeds and each individual animal. Currently, the most accurate method of predicting the parturition date is the prepartum progesterone drop, but the use of ultrasound parameters throughout gestation is still necessary to detect bitches before the onset of parturition [38]. An additional factor was the time that elapsed between neonatal extraction and the start of supportive measures, which was minimally prolonged due to obtaining samples for the study. Although this was minimal, it may have contributed to Apgar scores in highly sensitive, slightly immature EL-CS neonates. Our observation suggests that the appropriate timing of an EL-CS is of paramount importance. To minimize the vulnerability of EL-CS neonates in the immediate postpartum period and the demand for intensive neonatal resuscitation, EL-CS should ideally be performed after the onset of the first stage of parturition.

The expected difficulty in the noninvasive collection of the umbilical blood samples for lactate measurement, especially in small breed dogs [17], was confirmed in our patients. Whereas in a previous study at least 25 μL of umbilical blood was successfully collected from 70 of 94 puppies (74.5%) [17], in our study this was possible in 72 of 108 puppies (66.7%). This is less than mentioned by Groppetti et al. (2010), but in their population, only two out of 21 (9.6%) bitches belonged to small breeds [17] in contrast to 10 out of 22 parturitions (45.5%) in our study. In humans, umbilical blood gas analysis provides important information about the condition of the neonate [15]. In canine neonates, mixed acidosis was confirmed in the jugular venous blood sample 5 min after birth, and metabolic acidosis persisted until 1 h after birth when the second sample was collected [16]. We attempted to obtain 100 μL of blood for umbilical blood cord gas analysis but obtained only 55 complete samples in 108 neonates (51%), similar to Antończyk et al. [22], making this diagnostic tool unsuitable for clinical practice and requiring a learning curve. Further, the umbilical cord has a two-way direction of blood flow and with our technique, we were not able to ascertain that the taken blood sample originated from the neonate. This prompted us to investigate and compare the results of different parameters in fetal fluids other than the umbilical blood (amniotic fluid, urine) and to perform tests requiring less umbilical blood. We collected 87 (80.6%) amniotic fluid samples for the measurement of glucose, lactate, and cortisol concentration and 87 samples for urine cortisol concentration. The reason for the discrepancy in the number of samples collected and tests performed is due to the order and variety of tests that we performed at the beginning of the study.

The highest lactate concentration was found in VP puppies (in blood and amniotic fluid), and lower values were found in EM-CS and EL-CS puppies, which is in agreement with previous studies in dogs [18] and in humans [39]. Most likely, this is a consequence of uterine contractions during labor causing a physiological decrease in placental circulation with hyperlactatemia and acidosis at birth [40]. In another study, an initial lactate level of 6.7 mmol/L was reported in surviving neonates and 10 mmol/L in those who died in the first 24 h after VP [21]. In addition, a cut-off value for umbilical blood lactate of 5 mmol/L has been suggested to distinguish distressed from healthy puppies. Puppies that died within 48 h of birth had a mean lactate value of 12.2 mmol/L ± 6.7 SD [17]. Antończyk et al. [22] were able to confirm this correlation. In our study, umbilical lactate concentrations were available for three out of four puppies that died 4 days post parturition, and were 15.3, 23.3, and 20.1 mmol/L, respectively. Moreover, the duration of the active second stage of labor correlated significantly with the presence of fetal lactate [40]. In our study, the duration of the parturition was not always recorded, but puppies born at the end of the parturition did not have higher lactate concentrations in either umbilical blood or amniotic fluid. A very strong correlation between lactate concentrations in umbilical blood and amniotic fluid suggests that both fetal fluids can be used for the test, but higher concentrations in amniotic fluid must be considered. Lactate concentration did not correlate with relative growth rate except on day one.

In humans, higher blood glucose concentrations have been found in VP-delivered infants compared to EM-CS infants [41]. Similar to humans and in agreement with previous studies in dogs [12,18], significantly higher amniotic fluid glucose concentrations were found in VP puppies than in EM-CS puppies, but the concentrations reported by Groppetti et al. [18] were lower than in our study. On the contrary, Lucio et al. (2021) reported higher glucose concentrations in blood in fetal dystocia and EM-CS (after fetal or maternal dystocia) puppies when compared to eutocia puppies [19]. While some studies reported higher glucose concentrations in non-surviving compared to surviving neonates [21,22], others found that newborn puppies with glucose concentrations below 2.22 mmol/L usually had low Apgar scores and poor reflexes [12]. Balogh et al. (2018) showed that metabolites of carbohydrate and lipid metabolism in the bitch likely affect fetal concentrations and composition of fetal fluids [42], hence maternal hypoglycemia may affect neonates. Hypoglycemia is a reported syndrome affecting bitches presented for EM-CS and it is a particular risk in smaller dog breeds [42,43], probably due to consumption of glucose, which is reported to be normal or increased in at term dams and at the beginning of the dystocia [42,44]. These small EM-CS bitches often present after many hours in difficult labor as well as after or during episodes of dystocia where serum glucose is depleted. They are also often depressed and inappetent, which prevents them from correcting hypoglycemia. In contrast, one study reported that EM-CS dams had the highest glucose concentrations and the eutocia group had the lowest glucose concentrations, but the weight of the dams was not reported. In this study, a positive and significant correlation was found between maternal cortisol concentrations and glucose [19]. The reason for the lower glucose levels in EL-SC bitches could be that these bitches are often fasted before anesthesia, which would easily lead to lower serum glucose levels in their puppies. The EL-CS dams in our study were not fasted.

In our study, the umbilical blood glucose correlated moderately with amniotic fluid glucose concentrations and with Apgar scores at 5 and 15 min, but not with mortality in the first week. There was also no correlation with the growth rate of the puppies. These results suggest that neonatal hypoglycemia immediately after parturition is a common finding and should not be an immediate clinical concern. In addition, neonates in humans are reported to be much more resistant to hypoglycemia than adults [25]. These lower blood glucose levels early after birth have also been observed in dogs [45], foals [46], and calves [47], leading to the conclusion that they may represent an evolutionary adaptation to early life outside the womb.

In contrast to the previous results, where lower cortisol concentrations in puppies delivered by both types of CS were found [18], our results show the highest cortisol concentrations in puppies delivered by EM-CS and the lowest in those delivered by EL-CS. Again, this only partially supports data from human medicine indicating that vaginal delivery is more stressful for human neonates than CS. In humans, amniotic fluid cortisol concentrations increase abruptly during the last weeks of pregnancy due to activation of the hypothalamic–pituitary–adrenal axis, and the same can be expected in dogs [48]. Therefore, elevated cortisol levels are expected in at-term puppies (VP, EM-CS) compared to EL-CS puppies. Further stress to which the fetuses are subjected during dystocia results in higher cortisol levels in these puppies [19,49], suggesting that the stress of the CS procedure is indeed low. The study from Lucio et al. (2021) even showed that the type of dystocia influences the neonatal cortisol concentrations, with the highest concentrations in fetal dystocia puppies [19].

The correlation of cortisol concentrations in amniotic fluid and fetal urine is very strong, which is consistent with the results of a previous study, where concentrations in amniotic fluid and allantois were also strongly positively correlated. However, higher amniotic fluid but not allantoic cortisol concentrations were found in puppies not surviving 24 h after parturition [10] and lower concentrations were found in neonates with higher Apgar scores [49], which could not be confirmed in our study, because first, only one puppy died in the first 24 h and, second, puppies delivered by EL-CS had the lowest cortisol concentrations and the percentage of lower Apgar scores was the highest in this group. Puppies that died on the first, third, and fourth day (2 puppies) postpartum had amniotic cortisol concentrations of 13.3, 12.1, 2.0, and 9.2 mmol/L, respectively.

In our study, the amniotic fluid mean specific gravity in all parturition types was comparable to previous studies [50,51] which included CS births only. The utility of the amniotic fluid specific gravity as an indicator for neonatal survival was investigated by Fusi et al. who reported no significant differences in specific gravity of amniotic fluid in surviving and non-surviving puppies [51], which we can confirm.

There were no statistically significant differences among the three types of parturition regarding relative growth rate.

Because of various reasons, we had some missing values in the dataset, which is the main limitation of the study. Because of that, we were unable to establish or confirm already published cut-off values of cortisol, lactate, and glucose, which could be predictive of puppies’ survival. Further, because of the non-invasive nature of the study, we were possibly evaluating the maternal blood in contrast to the studies where the jugular blood of the neonates was examined. We are also aware that other factors such as breed, size, age of the dam, her vaccination, and health status, as well as low birthweight and litter size [5,6,13,14,32] may influence the survival of the neonates and these factors need to be included in the statistical analysis in the future studies.

## 5. Conclusions

Research on canine pregnancy and its effects on puppies’ vitality at birth, as well as a system for monitoring birth, are necessary to control the vitality of newborn dogs. The development of an easy-to-use monitoring system for at-risk puppies (e.g., orphaned, starved, low birthweight) would be desirable along with a simple Apgar evaluation for dog breeders to reduce the high neonatal mortality rate in kennels. It was shown here that measurement of urinary cortisol and amniotic fluid lactate are rapid tests that could serve as predictors of puppy vitality on the first day of life and help us identify puppies at risk. Our results showed that fetal and neonatal cortisol, lactate, and glucose concentration in puppies differed by the type of parturition. In summary, puppies born by EL-CS had the lowest lactate, glucose, and cortisol concentrations and the lowest Apgar score. However, the highest cortisol concentration was found in the EM-CS group, in which the Apgar score was lower than in the VP group, suggesting that optimal concentration rates for these three parameters should be determined to allow accurate assessment of neonatal vitality in dogs. Further studies are needed to determine whether these parameters can also be used as predictors of neonatal survival during the first weeks of life, when mortality is the highest.

## Figures and Tables

**Figure 1 animals-12-01247-f001:**
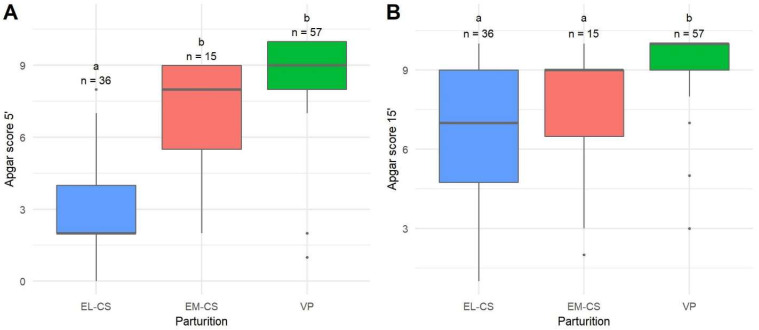
A comparison of Apgar scores 5 min (**A**) and 15 min (**B**) after parturition of puppies born by elective Cesarean section (EL-CS), emergency Cesarean section (EM-CS), and vaginal parturition (VP). Above each boxplot, statistically significant differences (*p* < 0.05) observed between parameters are marked with different lower-case letters, and the number of observations per group is indicated. The box represents the lower and upper quartiles, the line in the box denotes the median values, and whiskers the minimum and maximum values excluding outliers, which are marked with points.

**Figure 2 animals-12-01247-f002:**
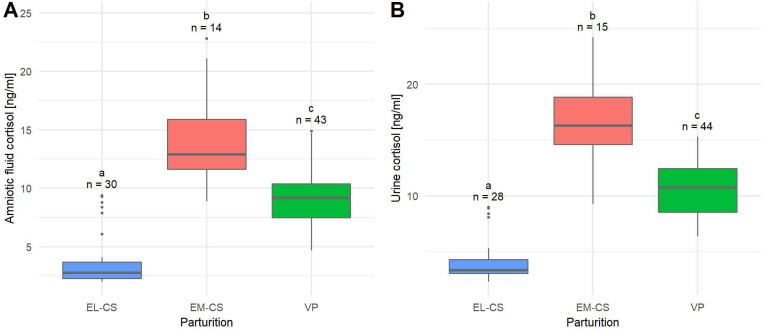
A comparison of amniotic fluid cortisol concentrations (**A**) and urine cortisol concentrations (**B**) in puppies born by elective Cesarean section (EL-CS), emergency Cesarean section (EM-CS), and vaginal parturition (VP). Above each boxplot, significant differences (*p* < 0.05) observed between parameters are marked with different lower-case letters, and the number of observations per group is indicated. The box represents the lower and upper quartiles, the line in the box denotes the median values, and whiskers the minimum and maximum values excluding outliers, which are marked with points.

**Figure 3 animals-12-01247-f003:**
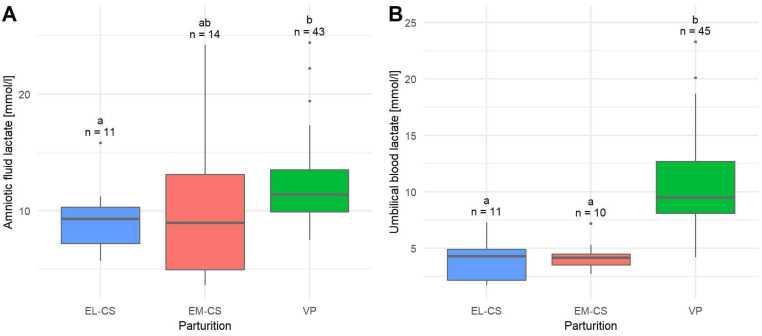
A comparison of amniotic fluid lactate concentrations (**A**) and umbilical blood lactate concentrations (**B**) in puppies born by elective Cesarean section (EL-CS), emergency Cesarean section (EM-CS), and vaginal parturition (VP). Above each boxplot, significant differences (*p* < 0.05) observed between parameters are marked with different lower-case letters, and the number of observations per group is indicated. The box represents the lower and upper quartiles, the line in the box denotes the median values, and whiskers the minimum and maximum values excluding outliers, which are marked with points.

**Figure 4 animals-12-01247-f004:**
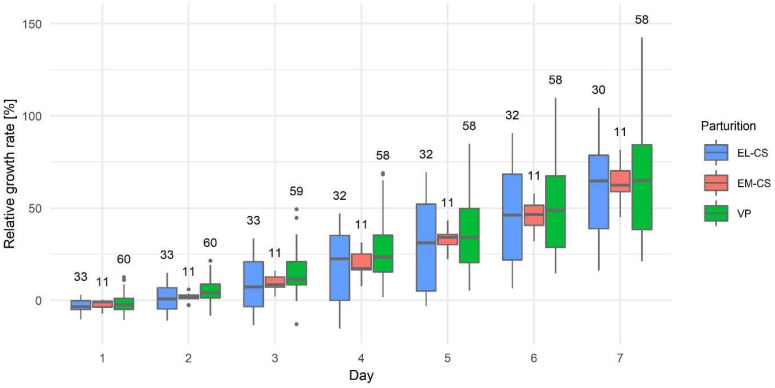
The puppies’ relative growth rate in the first week after parturition by elective Cesarean section (EL-CS), emergency Cesarean section (EM-CS), and vaginal parturition (VP). Above each boxplot, the number of observations per group is indicated. The box represents the lower and upper quartiles, the line in the box denotes the median values, and whiskers the minimum and maximum values excluding outliers, which are marked with points.

**Table 1 animals-12-01247-t001:** Number and proportion (%) of different breeds included in the study.

Breed Size	Breed	Number of Parturitions (%)	Age of Bitches (Years)	Number of Parity	Number of Female Puppies (%)	Number of Male Puppies (%)	Number of Puppies (%)
	Miniature Schnauzer	2 (9.1)	3–4	1–2	6 (4.9)	4 (3.2)	10 (8.1)
	Yorkshire Terrier	1 (4.5)	2.25	1	1 (0.8)	3 (2.4)	4 (3.3)
	Maltese	1 (4.5)	3.25	2	2 (1.6)	3 (2.4)	5 (4.1)
	Miniature Poodle	1 (4.5)	7	3	1 (0.8)	1 (0.8)	2 (1.6)
	Boston Terrier	4 (18.2)	2–5.5	1–3	11 (8.9)	5 (4.1)	16 (13)
	Jack Russell Terrier–Maltese mix	1 (4.5)	5	1	3 (2.4)	4 (3.3)	7 (5.7)
Total small breeds(<10 kg)		10 (45.4)			24 (19.5)	20 (16.3)	44 (35.8)
	Pembroke Welsh Corgi	2 (9.1)	2.5–2.75	1	3 (2.4)	7 (5.7)	10 (8.1)
	English Bulldog	1 (4.5)	2.5	1	2 (1.6)	2 (1.6)	4 (3.3)
	French Bulldog	2 (9.1)	2–2.5	1	5 (4.1)	8 (6.5)	13 (10.6)
	Beagle	1 (4.5)	4.25	3	3 (2.4)	4 (3.3)	7 (5.7)
	Dachshund	1 (4.5)	3	2	7 (5.7)	2 (1.6)	9 (7.3)
	Whippet	1 (4.5)	2.5	1	4 (3.3)	3 (2.4)	7 (7.3)
Total medium breeds (10–25 kg)		8 (36.4)			24 (19.5)	26 (21.1)	50 (40.6)
	Greater Swiss Mountain Dog	1 (4.5)	3.5	2	1 (0.8)	5 (4.1)	6 (4.9)
	Golden Retriever	1 (4.5)	2.5	1	5 (4.1)	6 (4.9)	11 (8.9)
	Labrador Retriever	1 (4.5)	2.75	2	8 (6.5)	2 (1.6)	10 (8.1)
	German Shepherd	1 (4.5)	3	2	1 (0.8)	1 (0.8)	2 (1.6)
Total large and giant breeds (>25 kg)		4 (18.2)			15 (12.2)	14 (11.4)	29 (23.6)

**Table 2 animals-12-01247-t002:** The modified Apgar scoring system used in this study [14,17,29].

Parameter	Points		
	0	1	2
Mucous membrane color	Pale or cyanotic	Pink	Reddish
Heart rate (bpm ^1^)	Absent or <120 bpm	120–180 bpm	>180 bpm
Respiratory rate	Absent or <6/min	Weak, irregular, 6–15/min	>15/min, rhythmic
Activity, muscle tone	Flaccid	Some flexions	Active motion
Reflexes irritability	Absent	Weak vocalization and weak reflex	Vigorous vocalization and immediate reflex

^1^ bpm: beats/min.

**Table 3 animals-12-01247-t003:** Number of puppies according to the type of parturition, sex, survival of newborns and the body weight at birth and on day 7.

Type of Parturition	Number of Puppies		Survival			Body Weight in Grams at Birth (Mean ± SEM)		Body Weight in Grams on Day 7 (Mean ± SEM)	
	Female	Male	Born Alive	Stillborn	Died after Discharge	Female	Male	Female	Male
VP	36	32	63	5	3	279 ± 12 (*n* = 33)	283 ± 17 (*n* = 28)	466 ± 23 (*n* = 31)	464 ± 32 (*n* = 27)
EL-CS	18	20	36	2	4	243 ± 16 (*n* = 17)	234 ± 18 (*n* = 19)	382 ± 32 (*n* = 14)	335 ± 32 (*n* = 16)
EM-CS	9	8	15	2	0	252 ± 48 (*n* = 8)	306 ± 49 (*n* = 7)	439 ± 64 (*n* = 5)	454 ± 47 (*n* = 6)

Legend: EL-CS: elective Cesarean section; EM-CS: emergency Cesarean section, VP: vaginal parturition; SEM: standard error of the mean.

**Table 4 animals-12-01247-t004:** Number and proportion of puppies according to Apgar scores 5, 15, and 60 min after parturition and type of parturition.

Type of Parturition	Apgar Score 0–3: Severe Distress (%)	Apgar Score 4–6: Moderate Distress (%)	Apgar Score 7–10: No Distress (%)
5 min after parturition			
VP	3 (5.3)	0 (0)	54 (94.7)
EM-CS	2 (13.3)	3 (20)	10 (66.7)
EL-CS	26 (72.2)	3 (8.3)	7 (19.4)
15 min after parturition			
VP	1 (1.8)	1 (1.8)	55 (96.5)
EM-CS	2 (13.3)	2 (13.3)	11 (73.3)
EL-CS	7 (19.4)	9 (25)	20 (55.6)
60 min after parturition			
VP	1 (1.7)	0 (0)	57 (98.3)
EM-CS	0 (0)	0 (0)	15 (100)
EL-CS	0 (0)	0 (0)	36 (100)

**Table 5 animals-12-01247-t005:** Number and proportion (according to the type of parturition) of the poorly, moderately, and adequately responsive puppies at 5, 15, and 60 min after parturition. Suckling, rooting, and righting reflexes were evaluated.

Type of Parturition	Number of Poorly Responsive Puppies (%)	Number of Moderately Responsive Puppies (%)	Number of Adequately Responsive Puppies (%)
Neonatal reflexes 5 min after parturition			
VP	3 (5.3)	13 (22.8)	41 (71.9)
EM-CS	8 (53.3)	2 (13.3)	5 (33.3)
EL-CS	28 (77.8)	3 (8.3)	5 (13.9)
Neonatal reflexes 15 min after parturition			
VP	2 (3.5)	3 (5.3)	52 (91.2)
EM-CS	5 (33.3)	2 (13.3)	8 (53.3)
EL-CS	19 (52.8)	12 (33.3)	5 (14.9)
Neonatal reflexes 60 min after parturition			
VP	1 (2)	1 (2)	56 (96)
EM-CS	0 (0)	1 (7)	14 (93)
EL-CS	0 (0)	3 (8)	33 (92)

**Table 6 animals-12-01247-t006:** Spearman’s correlation coefficient between measured parameters in the amniotic fluid, umbilical blood, urine, and relative growth rate.

	AM Glucose	AM Lactate	Apgar 5	Apgar 15	AM Cortisol	Cortisol in Urine	UB Glucose	UB Lactate
AM lactate	−0.1520	/						
Apgar 5	**0.4953 ***	−0.2141	/					
Apgar 15	0.3952	−0.1173	**0.7400 *****	/				
AM cortisol	0.2024	0.3513	0.0935	0.0376	/			
Cortisol in urine	0.1744	0.4820	0.1115	0.1087	**0.9199 *****	/		
UB glucose	**0.5307 ***	−0.1361	**0.5536 ****	**0.5854 ****	0.0680	0.0226	/	
UB lactate	0.0055	**0.8741 *****	−0.1602	−0.0013	0.4662	**0.5778 ****	−0.0404	/
Relative growth rate	0.2062	**−0.5378 ***	0.0422	0.0350	−0.2263	−0.3657	0.0901	**−0.5876 ****

Legend: AM—amniotic fluid, UB—umbilical blood, *p*-value < 0.001 (***), *p*-value < 0.01 (**), *p*-value < 0.05 (*), *p*-value < 0.1. Bold numbers indicate statistically significant results.

## Data Availability

Data are available from the authors upon request.

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
