# Peer review of "Canine Neonatal Assessment by Vitality Score, Amniotic Fluid, Urine, and Umbilical Cord Blood Analysis of Glucose, Lactate, and Cortisol: Possible Influence of Parturition Type?"

_animals, 2022, doi:10.3390/ani12101247_

Round 1

Reviewer 1 Report

I would like to thank the authors very much for their detailed responses and their thorough and thoughtful revision of this manuscript. I only have a few minor suggestions or comments.

Lines 111-118: Please rephrase this sentence to make it easier to read. Please also write out the parturition types before using the abbreviations. You may potentially say: “A thorough evaluation of non-invasive parameters and their correlation with neonatal vitality are still lacking in veterinary neonatology. Therefore, the aim of this study was to assess lactate and glucose concentrations in newborn umbilical blood and fetal amniotic fluid, cortisol concentrations in fetal amniotic fluid and urine of newborn puppies, and to evaluate the association between these biomarkers with newborn vitality score, parturition type (VP versus EL-CS versus EM-CS) and puppy survival within the first seven days of life. We also aimed to investigate the association between parturition type and puppy survival and growth.”

Table 3 and line 463: I suggest using body weight instead of mass in reference to puppy weights.

Line 359: Please add “P” when referencing the P value.

Line 422-424: The authors write that “The difference in the amniotic fluid glucose concentrations between parturition types was not significant (p = 0.0666) (Figure 3)”, but Figure 3 still depicts significant difference between the parturition types; please revise. The authors may consider removing Figure 3 and leave the description in the text on amniotic fluid glucose similar to umbilical blood glucose, or add umbilical blood glucose to Figure 3 if they want to present these results even if no significant differences were detected.

Specific gravity of canine fetal fluids has been described before; please see these previous studies, which can be compared to the authors' results:

  • Balogh O, Roch M, Keller S, Michel E, Reichler IM. The use of semi-quantitative tests at Cesarean section delivery for the differentiation of canine fetal fluids from maternal urine on the basis of biochemical characteristics. 2017 Jan 15;88:174-182. doi: 10.1016/j.theriogenology.2016.09.024.
  • Fusi J, Bolis B, Probo M, Faustini M, Carluccio A, Veronesi MC. Clinical Trial on the Usefulness of On-Site Evaluation of Canine Fetal Fluids by Reagent Test Strip in Puppies at Elective Caesarean Section. Biology (Basel). 2021 Dec 27;11(1):38. doi: 10.3390/biology11010038

Correlations, Section 3.5, lines 480-498: I understand the authors’ point however, I still think Apgar 60 should be mentioned here. If the authors do not want to include it in the Table, then please include it in the text, similarly how you described the correlations of growth rate on day 2 and later. It is important to add Apgar 60 due to its relevance to overall puppy prognosis, even if it has no correlations with your biomarkers.

Author Response

I would like to thank the authors very much for their detailed responses and their thorough and thoughtful revision of this manuscript. I only have a few minor suggestions or comments.

Dear Reviewer!

Thank you for your kind words. We very much appreciated the encouraging and helpful comments. We strongly believe that the comments and suggestions have increased the scientific value of revised manuscript by many folds. We have taken them fully into account in revision. We are submitting the corrected manuscript with the suggestion incorporated the manuscript. The manuscript has been revised as per the comments given by the reviewer, and our responses to all the comments are as follows:

Lines 111-118: Please rephrase this sentence to make it easier to read. Please also write out the parturition types before using the abbreviations. You may potentially say: “A thorough evaluation of non-invasive parameters and their correlation with neonatal vitality are still lacking in veterinary neonatology. Therefore, the aim of this study was to assess lactate and glucose concentrations in newborn umbilical blood and fetal amniotic fluid, cortisol concentrations in fetal amniotic fluid and urine of newborn puppies, and to evaluate the association between these biomarkers with newborn vitality score, parturition type (VP versus EL-CS versus EM-CS) and puppy survival within the first seven days of life. We also aimed to investigate the association between parturition type and puppy survival and growth.”

Thank you for your suggestion. The suggested sentence is now included in the study.  Lines 111-119

Table 3 and line 463: I suggest using body weight instead of mass in reference to puppy weights.

Mass was changed into body weight throughout the manuscript.

Line 359: Please add “P” when referencing the P value.

Thank you for noticing missing P. It is now added. 

Line 422-424: The authors write that “The difference in the amniotic fluid glucose concentrations between parturition types was not significant (p = 0.0666) (Figure 3)”, but Figure 3 still depicts significant difference between the parturition types; please revise. The authors may consider removing Figure 3 and leave the description in the text on amniotic fluid glucose similar to umbilical blood glucose, or add umbilical blood glucose to Figure 3 if they want to present these results even if no significant differences were detected.

Thank you very much for noticing this. We have corrected the mistake. We have deleted the Figure 3, the number of collected samples was added in the text.

Specific gravity of canine fetal fluids has been described before; please see these previous studies, which can be compared to the authors' results:

  • Balogh O, Roch M, Keller S, Michel E, Reichler IM. The use of semi-quantitative tests at Cesarean section delivery for the differentiation of canine fetal fluids from maternal urine on the basis of biochemical characteristics. 2017 Jan 15;88:174-182. doi: 10.1016/j.theriogenology.2016.09.024.
  • Fusi J, Bolis B, Probo M, Faustini M, Carluccio A, Veronesi MC. Clinical Trial on the Usefulness of On-Site Evaluation of Canine Fetal Fluids by Reagent Test Strip in Puppies at Elective Caesarean Section. Biology (Basel). 2021 Dec 27;11(1):38. doi: 10.3390/biology11010038

Thank you for pointing out the literature on specific gravity of amniotic fluid in puppies. We have included the suggested literature in the discussion section of the revised version of our manuscript. (Lines 645-649).

At the same time, we were reviewing the data and found one false value in the data (specific gravity was written 0.022 instead of 1.022), which we corrected and hence we corrected the results as well.

Correlations, Section 3.5, lines 480-498: I understand the authors’ point however, I still think Apgar 60 should be mentioned here. If the authors do not want to include it in the Table, then please include it in the text, similarly how you described the correlations of growth rate on day 2 and later. It is important to add Apgar 60 due to its relevance to overall puppy prognosis, even if it has no correlations with your biomarkers.

As suggested, we included the information that there were no correlation between Apgar 60 and other variables in the text – line: 492.

Reviewer 2 Report

This study assesses non-invasive methods, e.g. Apgar score, concentrations of glucose, lactate and cortisol, for pup neonatal assessment, and the influence of parturition type on these non-invasive methods, survival and pup growth. The reasoning for the study is well introduced along with the background literature. The methods are significantly improved, however the statistical analyses are my main area of concern. The results section is improved, however i) none of results include the test statistics and the degrees of freedom, ii) at least 1 non-parametric test is missing, and iii) I do not understand why multivariate analyses are not possible (sharing the data with me and the attempted models’ outputs would be helpful). Several of my major comments from my first revision were insufficiently account for in the revised manuscript and rebuttal. The statistical analyses are the major weakness of this manuscript.  If the authors can share the data and the attempted model outputs, I would be please to review the revised version of this manuscript. Based on what the authors wrote in the rebuttal and revised manuscript about the data and attempted model outputs, I do not know how this will affect the results, discussion and conclusions. I cannot recommend that the manuscript be currently accepted, however I recommend major revisions.

The authors revised the manuscript and accounted for several of my comments but not all of my major comments.  A few of my major comments were not accounted for or were insufficiently accounted for. There are few line numbers in the rebuttal to my major comments for the abstract (note: my major comment for the abstract spanned the abstract, introduction and discussion) and introduction making it difficult to review the revised ms. I would prefer to have line numbers to easily find revised sections accounting for my comments rather than searching the entire revised manuscript for the revisions that account for my comments. There was no rebuttal for my second major comment, however the last paragraph of the introduction was revised. There is a short rebuttal to my third comment without line numbers “Introduction section was changed according to your suggestions and studies that were done on puppies were included into the introduction. Please look at the introduction section.”.  

Major comment 1: With regards to my first major comment from my first review, I explained how “estimates of the hormone metabolites in the faeces of animals vary by sex, diet of individuals, temperature, season and bacteria, to name some of the variables affecting the estimation of hormone metabolites (Goymann 2012 https://doi.org/10.1111/j.2041-210X.2012.00203.x).”  I also wrote the following: “Have previous experiments been done to conclude that pup urine cortisol concentration is a reliable measure of the physiological status of an individual?  There is no mention of these matters in the introduction, methods and discussion. If yes, please describe how the different variables affect metabolite concentrations and how you accounted for them to generate the reliable measure of the physiological status of a pup in the ms (e.g. introduction, methods and discussion).  If no, the authors should i) make a statement that is more modest regarding the non-invasive cortisol urine concentration method, and ii) describe this in the introduction, methods and discussion.”  I found no sentences in the revised manuscript that describe the reliability of hormone metabolites in the faeces.  In the rebuttal, the authors wrote: “To our knowledge there is a paucity of information regarding values and reliability of urine cortisol determination in neonate puppies.”  The authors wrote the following in their rebuttal: “As well, we wanted to find clinically useful parameters that clinicians can use at the time of birth”. With this goal in mind, I believe that my first major comment from my first review remains valid for clinicians.  The literature or lack of literature about the reliability of hormone metabolites in faeces should be described in the manuscript. The reliability of hormone metabolites in faeces on the response variables of interest should be described in the manuscript.  If clinicians should use the findings in this manuscript, they should also understand the reliability of hormone metabolites in faeces on the response variables of interest. For example, Goyman (2012) provides examples of how the faecal hormone metabolite concentrations can be similar with the blood hormone concentrations but actually differ because of other variables. As such, the results in Figure 2B may be misleading, and this should be discussed. If this were done by the authors in the introduction and the discussion, I would consider this comment as completed.

Major comment 2: My major concern with the manuscript (originally submitted and revised) focused on the data analyses, and I wrote extensively about this in my first review. The authors did not provide codes or outputs of the models they attempted to run. The authors also wrote: "However, it turned out that due to lack of data, none of the simpler models that met the model assumptions provided additional information that has not yet been included in the manuscript (the one suitable model was not significant) and most of them were not suitable for the data and even after transformations of the variables, the model fit was not satisfactory." I would have preferred to read the outputs, especially since I wrote a lengthy major comment about the statistical methods. There are many non-parametric tests and Spearman correlation coefficients, and none of these can account for the variance of other variables.

I would like the authors to share the data with me, the model outputs of the models they attempted to run (along with the types of models they attempted to run in association with the model outputs). I proposed some models in my first review, and I still wonder about them. Some of the replies to my major comments about the statistical analyses were insufficient, and it still seems that the authors may have the data to perform some of these models (see below for some examples). I also think that the data should shared with the readers, so clinicians can have access to the data and learn from the data in order to improve non-invasive methods at the time of birth.

The authors replied: “For various reasons, we have some missing values in the dataset (all data are recorded only for 36 pups, most were delivered vaginally, 2 with elective CS and 2 with urgent CS)”. Yet, Table 3, reports many more elective CS and urgent CS.  I understand that you have all the data for only 2 pups delivered by elective CS and only 2 pups delivered with urgent CS, but you report summary data in Tables 1 and 3. Why can’t the data in Tables 1 and 3 be used to run more than non-parametric tests and bivariate correlation tests, such as generalized linear (mixed) models? If some variables have extensive data missing but others do not, this should not invalidate the possibility of running models with the variables that have sufficient data.

The authors replied: “However, it turned out that due to lack of data, none of the simpler models that met the model assumptions provided additional information that has not yet been included in the manuscript (the one suitable model was not significant) and most of them were not suitable for the data and even after transformations of the variables, the model fit was not satisfactory.”  I would like to see the model outputs.

I also appreciate the clarifications and the stated limitations of the data (e.g. in the discussion on 652-660) in the revised manuscript with regards to the data.

The following was one of my previous comment, and the authors did not provide a reply: “How many pups in the study have the same mother and were born in the same litter?  Were mothers repeatedly sampled between litters?  If the answer is greater than once, you should account for pseudoreplication due to repeated measures from the same mother.” This affects the non-parametric models and bivariate correlations, which assume independence of the data.

Lines 371-378: “In the EL-CS group 26 puppies (72%) were in severe distress 5 min after parturition, whereas 54 (95%) and 10 (67%) of puppies born with either VP or EM-CS showed no distress. Differences in the distribution of Apgar scores were also observed 15 min after parturition, with the distribution of the puppies born with EM-CS and VP similar to 5 min after parturition. However, most puppies born with EL-CS were not in distress and the proportion of severely distressed puppies born with EL CS decreased compared with Apgar scores 5 min after parturition. As a result, the proportion of puppies being moderately distressed 15 min after parturition increased compared with the Apgar score 5 min after parturition.” This is descriptive, and statistical results would improve these results. For example, the proportion of puppies being moderately distressed at 15 min compared to at 5 min could be assessed.

The test statistics and degrees of freedom for each result need to be reported. Only the p values are reported, and this is insufficient. Table 6 needs to include the degrees of freedom, so readers can figure out the sample size for each of the correlations. As an added bonus, the inclusion of the test statistics and degrees of freedom to your manuscript may help your manuscript’s results be included in future meta-analyses.

Given the information in Table 1 and Table 3, why can an event history analysis (semi-parametric or parametric model) not be performed to assess the survival of pups?  In the rebuttal to my 14th major comment in my previous review, the authors wrote: “The problem here is that our mortality rate was really low because we took care of the puppies immediately when we noticed a problem. Normally, the mortality rate is higher. Therefore, this was included in the conclusion: “As stated above, we could not assess in this study whether these parameters could be predictors of neonatal survival due to lack of data. However, we think that our data suggest that this might be the case, but further studies are needed.”” I do not understand this reply. Survival was high, but this does not stop an event history analysis from being performed.

Based on Figure 5, the relative growth rate data seems like a generalized linear mixed model could be used to assess the effect of birth type (fixed effect) on the relative growth rate (e.g. one model with the absolute change in mass and one model with the percent change in mass) while controlling for race. The identity of the pups could be included as random effect. 

Based on what the authors wrote in the rebuttal and revised manuscript about i) the data, ii) the attempted model outputs, iii) the apparent availability of data in Tables 1 and 3, and iv) the statement about a non-significant model that was not described, I do not know how this affects the results, discussion and conclusions. 

Minor comments:

Regarding my second major comment from my first revision, I come from the field of evolutionary ecology and behavioural ecology. In this field, we write what hypothesis is under investigation, and we write predictions.  I will not impose my field’s methods on this manuscript.  I do think the manuscript would be improved with clearly-stated predictions. I have read studies by veterinarians with clearly stated hypothesis and with expected outcomes.  Your study may have so many predictions that readers may confused. The aims of the study were improved, and the sentence is clearer for readers (lines 111-118). I would have preferred a rebuttal to this major comment with line numbers.

The introduction non-invasive methods for pup neonatal assessments was revised, and I believe this improves the manuscript by introducing past research for readers.  My third major comment from my first revision was completed. 

The sentence on line 525 is a stand alone paragraph.  Please correct this.

Sincerely,

The Reviewer

Author Response

Thank you kindly for your opinion. Since this is a veterinarian study and a clinical study all of the requests are just not possible to be included into the consideration. We tried our best to include as many suggestions as it was possible and we have made a lot of changes already.

We hope the new version will be suitable for you.

Kind regards

Reviewer 3 Report

The manuscript has improved considerably. The authors should be commended for their efforts to improve the manuscript. Authors have made changes in the Materials and Methods section to provide more information on the experimental design. More importantly, authors have mentioned and thoroughly discussed the important scientific flaws of the present research. Additionally, substantial modifications have been performed throughout.

Further comments are as follow:

Title

Please, be more specific on what you have analyzed (glucose, cortisol and lactate), in order to include such evaluation in the overall clinical assessment of a neonate puppy.  

Suggested title: “Canine neonatal assessment by vitality score, amniotic fluid, urine and umbilical cord blood analysis of glucose, lactate and cortisol: possible influence of parturition type?”

Introduction

Lines 51-54: this paragraph is disconnected from the main idea. Please, make a tighter link with the former information.

Line 75: I think your experiment was not designed to identify a biomarker of neonatal vitality, since the vitality score is the most recognized manner to do so. I would suggest changing the term “biomarker”.

Line 85: tissue hypoperfusion is not the only cause of anaerobic cellular metabolism, as hypoxemia (low pO2 concentration by pulmonary oxygenation deficiency) can play a significant role on tissue metabolism.

Materials and Methods

Line 309: it is preferable to use the standard error of the mean (SEM), which is more feasible for a true population mean. The SEM is always smaller than the SD.

Discussion

Lines 552: this highlights the special attention we have to draw on timing c-section. Veterinarians should prioritize operating after the onset of first stage of whelping. My suggestion is to make this point clearer in your discussion.

Author Response

The manuscript has improved considerably. The authors should be commended for their efforts to improve the manuscript. Authors have made changes in the Materials and Methods section to provide more information on the experimental design. More importantly, authors have mentioned and thoroughly discussed the important scientific flaws of the present research. Additionally, substantial modifications have been performed throughout.

Thank you so much for your kind words. We appreciate the time and effort that you have dedicated to providing your valuable feedback on our manuscript. We are grateful for your helpful comments, and we have been able to incorporate changes the revised manuscript.

Further comments are as follow:

Title

Please, be more specific on what you have analyzed (glucose, cortisol and lactate), in order to include such evaluation in the overall clinical assessment of a neonate puppy.  

Suggested title: “Canine neonatal assessment by vitality score, amniotic fluid, urine and umbilical cord blood analysis of glucose, lactate and cortisol: possible influence of parturition type?”

Thank you for this suggestion. The suggested title is now used in the revised manuscript.

Introduction

Lines 51-54: this paragraph is disconnected from the main idea. Please, make a tighter link with the former information.

Thank you for pointing this out. We connected it with a former information. (Lines 50-55)

Line 75: I think your experiment was not designed to identify a biomarker of neonatal vitality, since the vitality score is the most recognized manner to do so. I would suggest changing the term “biomarker”.

Thank you for your explanation why we should change biomarker with other term. We changed it to indicators. I hope you agree with this term. (Line 75)

Line 85: tissue hypoperfusion is not the only cause of anaerobic cellular metabolism, as hypoxemia (low pO2 concentration by pulmonary oxygenation deficiency) can play a significant role on tissue metabolism.

Thank you for pointing this out. We agree with this comment. Therefore, we have incorporated this information into the introduction section (line 85).

Materials and Methods

Line 309: it is preferable to use the standard error of the mean (SEM), which is more feasible for a true population mean. The SEM is always smaller than the SD.

We changed SD to SEM. The changes were made throughout the results section and some adjustments were needed in introduction, materials and methods, and discussion (lines: 93, 130, 135, 275, 289, 311, 591).

Discussion

Lines 552: this highlights the special attention we have to draw on timing c-section. Veterinarians should prioritize operating after the onset of first stage of whelping. My suggestion is to make this point clearer in your discussion.

Thank you so much for your comment. You have raised an important point here and we incorporated this aspect into the revised manuscript (Lines 544-547).

This manuscript is a resubmission of an earlier submission. The following is a list of the peer review reports and author responses from that submission.

Round 1

Reviewer 1 Report

The present manuscript aims to propose a predictive analysis of neonatal outcome during the first month of life, based on vitality score, lactate, glucose and cortisol levels at birth, taking into account the birth mode. For this purpose, neonates were classified according to their vitality score at the immediate postpartum and prospective correlations with amniotic fluid, blood umbilical cord and urine metabolic variables were evaluated. The mortality rate during the neonatal period was also considered.

This manuscript is interesting and, in fact, this is an important research area and studies on canine neonatology have to be stimulated for an overall understanding of neonatal physiology. The study on puppies’ assessment as a clinical parameter in neonatology should be encouraged. Despite the interesting area of experimentation, there are important issues that derail its publication in the present form. The material and methods section lacks some essential information.

My most important concern relates to the umbilical blood itself. While hemodynamically active, umbilical vessels blood content is of both maternal and fetal origin, unless you have previously isolated the umbilical vein. Therefore, you cannot affirm that the metabolic analysis you are performing is restricted to the neonate during the transition phase. You are possibly analyzing fetal blood mixed with maternal blood under the intra-uterine environment. Thus, the proposing analysis for the main goal of predicting short-term neonatal outcome is biased.

Additionally, you included the birth mode in your statistical model, but you have neglected the data in the discussion. Is it possible to stablish specific predictive neonatal variables according to the obstetrical condition at birth? I believe that this is the main scientific question you are trying to answer. However, it remained unexplained.   

Some of the major comments and criticisms include:

Title:

  • Authors should refer to the birth mode. The title does not reflect the purpose of this study: influence of birth mode on neonatal outcome.
  • In addition, the title is award. How do you expect to evaluate puppy growth by analyzing amniotic fluid and umbilical cord? Please clarify.
  • The "Apgar score" was first applied strictly to human babies. I suggest referring to a more general evaluation: vitality score.

Simple Summary

Line 23: You cannot affirm that birth mode has no impact on puppies’ survival as you did guarantee neonatal assistance to control any abnormality that could have influenced mortality. Please, rewrite.

Abstract

Line 37-38: Your conclusion is not paired with your objectives. Please, reformulate the conclusion in order to answer the scientific question established at the objective.

Introduction:

Although the Introduction is well written, authors should try to address thoroughly the influence of birth mode on neonatal morbidity and mortality.

Line 47: Are you referring to neonatal mortality in dogs? Please, specify the animal species.

Line 49: please suppress the information “Guide dogs for the blind”, as it is not relevant for the data you are showing.

Line 69-71: these markers are strictly for metabolic issues and does not indicate any respiratory impairment or acid-base imbalance. Thus, the veterinary assistance will be only in a metabolic point of view.

Lines 84-89: your objectives are confuse. Reading the Introduction it sounds as that the type of whelping is your main goal of study. On the other hand, the objective are designed to answer another scientific question. Please, be consistent with your ideas.

Material and methods:

You should place the full description of the study population in the MM section. Some very important information should be provided:

  • How many puppies per bitch?
  • Please describe in detail the neonatal management: did the authors subject puppies from all experimental groups to any stimulation procedure after birth? Please, detail what procedures you performed for neonatal resuscitation: manual ventilation, mechanical ventilation, exogenous oxygen, etc...
  • Please, describe in detail how did you sampled the umbilical cord: by venous puncture (which needle caliper?) or by manual ejection. Also, provide more information on urine sampling.
  • Another important information is what vessel direction the umbilical blood was collected? The umbilical cord has a two-way direction of blood flow, i.e., from maternal or fetal origin. Please clarify if the blood was actually from the fetus / neonate.

Line 120: Did you give any premedication?

Lines 156-158: this procedure may have biased the evaluation of neonatal weight gain.

Line 189: how did you measure specific gravity in the amniotic fluid?

Lines 189-190: Did you previously validate the glucose and lactate assay for the amniotic fluid?

Line 197-199: Please, establish a more specific time range for urine sampling. Have the cortisol assay been previously validated for urine samples?

Results:
I think that all these basic information of the study population should be provided in the Material and Methods.

Authors should clarify the main reasons for performing EM-CS in five bitches. Were the bitches in fetal or maternal dystocia? How many bitches were in uterine atony? Neonatal outcome depends directly on the type of dystocia.

Figure 12: please, explain the symbols 

Discussion

Line 407: “fetal and neonatal (urine) fluids”

Line 421: However, it is of utmost importance to define low birth weight puppies for determined breed sized bitches.

Lines 445-447: unless CS is performed during the first stage of labor (before rupture of the allantoic sac), puppies will be less prone to making an uneventful fetal-to-neonatal transition. This relates to the onset of adaptive mechanisms that are triggered during whelping.

Lines 454-456: but this management was the same employed for the EM-CS. Thus, this does not explain the difference in vitality score.

Lines 497-499: this is not the case for Lucio et al. (2021), considering EM-CS according to the type of dystocia. Please, see: Lúcio CF, Silva LCG, Vannucchi CI. Perinatal cortisol and blood glucose concentrations in bitches and neonatal puppies: effects of mode of whelping. Domest Anim Endocrinol. 2021 Jan;74:106483. doi: 10.1016/j.domaniend.2020.106483.

Lines 503-504: according to Lucio et al. (2021), hypoglycemia occurs in both vaginal and EM-CS whelping bitches.

Lines 509: this is not the case for Lucio et al. (2021)

Lines 520-521: according to Lucio et al. (2021), the level of neonatal cortisol at birth depends of the type of dystocia, being higher only in fetal dystocia puppies.

Conclusion

You should consider making conclusions on the comparison among birth mode.

Reviewer 2 Report

This study assesses non-invasive methods, e.g. Apgar score, concentrations of glucose, lactate and cortisol, for pup neonatal assessment, and the influence of parturition type on these non-invasive methods, survival and pup growth. The abstract is clear, however the authors make a few claim that are insufficiently modest with regards to the reliability of the non-invasive cortisol urine concentration method. The reasoning for the study is well introduced, however there are no hypotheses and predictions.  The predicted associations and potentially interactions are not stated. Tables 1 and 2 are useful methodological keys.  I suggest an event history analysis to assess pup survival and generalized linear mixed models to assess pup growth and lactate, glucose and cortisol concentrations. Such analyses would control for other fixed effects (i.e. independent variables in mixed models). Some of the analyses that the authors state should be studied in a further research projects appear to be possible to assess within this manuscript within 1 model, (e.g. the influence of neonatal glucose, lactate and cortisol concentrations on pup survival over the first weeks after parturition; what affects apgar scores). The results and discussion may (i.e. would likely) change if non-parametric analyses were replaced by mixed models, and these models would also account for pseudoreplication due to repeated measures per individual. I have several major comments. I cannot recommend that the manuscript be currently accepted, however I recommend major revisions. If the statistical analyses are significantly revised, I would be please to review the revised version of this manuscript.

Major comments:

Abstract

1) Lines 37-38: “Non-invasive analysis of puppies’ fluids can serve as a good predictor of neonatal stress.” A similar sentence on lines 545-547: “Measurement of urinary cortisol and amniotic fluid lactate are rapid tests that could serve as good predictors of puppy vitality on the first day of life.” Hormone concentrations from feces are based on hormone metabolites in the faeces.  Hormones are metabolized, and the metabolites are excreted in the feces.  Estimates of the hormone metabolites in the faeces of animals vary by sex, diet of individuals, temperature, season and bacteria, to name some of the variables affecting the estimation of hormone metabolites (Goymann 2012 https://doi.org/10.1111/j.2041-210X.2012.00203.x).  The sexes often metabolize the same hormone differently, such that intersex comparisons of hormone metabolite concentrations may not be possible.  Do male and female dogs metabolize cortisol differently?  Diet and differences in diet between animals can alter how hormones are metabolized. On lines 95-96: “All bitches were fed only FDA approved food for pregnant dogs.” Changes in temperature (seasonal, daily) can alter how hormones are metabolized.  If a pup urinated a larger volume of urine than another pup, the concentration of cortisol metabolites may be low in the pups with a larger urine volume than in pups with a lower urine volume due to a dilution effect. Differences between individuals in bacterial gut composition can alter how hormones are metabolized.  This increases the random variance and may lead to systematic noise, which can significantly distort the hormonal status of an individual in a non-random way.  To validate if the non-invasive analysis of pup urine cortisol concentration is a reliable measure of the physiological status of an individual, many experiments are required.  Have previous experiments been done to conclude that pup urine cortisol concentration is a reliable measure of the physiological status of an individual?  There is no mention of these matters in the introduction, methods and discussion. If yes, please describe how the different variables affect metabolite concentrations and how you accounted for them to generate the reliable measure of the physiological status of a pup in the ms (e.g. introduction, methods and discussion).  If no, the authors should i) make a statement that is more modest regarding the non-invasive cortisol urine concentration method, and ii) describe this in the introduction, methods and discussion. Although, figures 7 and 8 show similar patterns between cortisol concentrations in amniotic fluid and urine, i) the models from which these results are based on do not control for other independent variables that can affect the response, and ii) the same pattern could be due to any of the variables mentioned above cancelling each other. 

Introduction

2) Although I understand the questions that the authors are interested in, the authors do not introduce: i) a hypothesis, ii) predictions to assess this hypothesis, and iii) the association between parturition type and survival and growth is not clearly stated.  If I only read the introduction, without reading the abstract, I would not understand that the authors are interested in estimating the association between parturition type and survival and between parturition type and growth.  Lines 84-89: “The aim of this study was to assess the concentration of lactate and glucose in the umbilical blood and amniotic fluid, to assess the concentration of cortisol in the amniotic fluid and urine of the newborn puppies, and to assess the correlation between these values with survival within first seven days of life. Viability of the newborn puppies was assessed using the modified Apgar scoring for dogs. The type of parturition (VP versus EL-CS versus EM-CS) was also considered.”  From the introduction, I would assume that the authors might predict, for example, that an increase in lactate concentration in the amniotic fluid and in the umbilical blood decreases pup survival, or that an increase in lactate concentration in the amniotic fluid and in the umbilical blood increases the instantaneous risk of pup death.

3) The authors are interested in testing if non-invasive methods can reliably be used for pup neonatal assessment.  I suggest that the manuscript would be improved if the authors further developed on this in the introduction and further explained what is known and we need to know. For example, the association between pup cortisol concentration in allantoic and amniotic fluids and survival has already been studied, and the association between pup cortisol concentration in allantoic and amniotic fluids and the Apgar score has already been studied (Bolis et al. 2017). Is the pup urine cortisol metabolite concentration measure faster and more reliable (as described above, it is likely not mor reliable)? How might the pup urine cortisol metabolite concentration estimates improve pup neonatal assessments? 

Have no other studies assessed non-invasive methods for pup neonatal assessments (e.g. Apgar score (only Veroni 2016 assessed the Apgar score for pup neonatal assessments?; Bolis et al. 2017),   If so, introduce this further in the introduction, i.e. more than lines 59-63.  If others have assessed this, introduce this further and add references.  In the discussion, the authors write on lines 407-409 : “The clinical parameters assessed in this study were selected because they are widely used in veterinary medicine for subjective assessment of puppies to determine their viability.”

Methods

4) I am not an expert in veterinarian science. I am an ecologist. I have administered drugs for anaesthesia, analgesia and euthanasia following an animal research license. As such, I do not have the expertise to assess the scientific merit of this guidelines used by the authors for the medical management of parturition, e.g. section 2.2.  

5) Lines 99-100: “Each newborn puppy was weighed, assessed using the modified Apgar scoring system …” Add citations for the modified Apgar scoring system.

Each newborn puppy was weighed, and pups were weighed everyday for the first week and once per week until week 8.  Please report who recorded the mass, the precision of the scale(s), the mean mass and standard error of the pups at birth and the end of the study for male and female offspring.

6) The methods for lactate, glucose and cortisol concentrations are insufficiently described.

7) Section 2.7 Statistical analyses. The statistical analyses are non-parametric tests, e.g. Spearman rank correlation coefficient, non-parametric Wilcoxon rank sum test or Kruskal-Wallis rank sum test. I think that I can help improve the manuscript results with the following suggestions.  There are 123 pups and 20 of mothers.  None of the models control for other variables’ influence on the variance of the response variable.  Many of the results could change with the models that I propose, and this could affect the interpretation of the results and the discussion.

How many pups in the study have the same mother and were born in the same litter?  Were mothers repeatedly sampled between litters?  If the answer is greater than once, you should account for pseudoreplication due to repeated measures from the same mother.  For pup survival, the state (alive or dead) of the pups was measured multiple times, and you should account for pseudoreplication due to repeated measures of the same pups.  For pup growth, you should account for pseudoreplication due to repeated measures of the same pups.  This requires mixed models. 

For the survival analysis, I suggest an event history analysis.  If you need to learn to run such models, I co-authored an open-access introduction to event history analyses with definitions of terms and the 4 functions, a decision tree, and 4 example data sets with associated R scripts to help researchers learn how to use these models (Landes et al. 2020 https://doi.org/10.1002/ecs2.3238).  To account for pseudoreplication due to repeated measures of the pup state over time, you should include frailty.  I assume you know the state of each pup at the end of the study, which is nice information to have.  The survival R package and its vignettes by Therneau are quite useful for organizing the repeated measures data, running an event history analysis, e.g. semi-parametric Cox proportional model with frailty, and assessing the proportional hazard assumption of a semi-parametric Cox proportional model.  If you have multiple pups born from the same mother, you can account for random effects for the mothers and for the pups due to repeated measures.

Please note that I am only provided potential models with fixed and random effects that you may find of interest. I am not attempting to dictate your models: I am simply trying to give you examples of how to improve your manuscript’s statistical analyses and results, which may affect your discussion.  I will likely ask you why you add specific variables in a future review, so it would be good if in a revised version of the manuscript and/or in your rebuttal, you clearly explained why some variables were included and others were not.  

Litter size affected pup survival (Moon et al. 2000 https://doi.org/10.5326/15473317-36-4-359).  Why was litter size not included as a continuous variable in the model to assess pup survival?  Does the age and the experience (1st parturition, 2nd, 3rd, 4th, etc. or dichotomous, i.e. 1st vs > 1st parturition) of the mother affect pup survival?  Does pup sex affect pup survival?  In humans and non-human animals, these variables can affect offspring survival.  In the event history analysis, survival over time (with date of event, i.e. death) can be assessed as a function of fixed effects, e.g. you can include the type of parturition, breed, age and sex of pups, age, mass and experience of mothers, litter size per parturition, apgar score, lactate, glucose and cortisol concentrations in 1 model.  You can also account for interaction effects that you think are important (e.g. breed and another variable). You can include pups and mothers as random effects. You could include the 7 pups that died after discharge in the analysis. You can test for multicollinearity.  You can use the multicomp R package’s ghlt function to assess post-analysis comparisons.

For pup growth, a Gaussian distribution is likely good (to be assessed with the model residuals), you can include the type of parturition, breed, age, sex and birth mass of pups, age, mass and experience of mothers, litter size per parturition, apgar score, lactate, glucose and cortisol concentrations in 1 generalized linear mixed model with a Gaussian distribution and random effects for repeated pup and mother (e.g. multiple pups in the data share the same mother) measures. You can also account for interaction effects that you think are important (e.g. breed and another variable). If you run a model with the change in mass, e.g. percentage of mass change, you cannot include mass at birth of pups as a fixed effect, since birth mass is in the percent mass change. Did the lactate and glucose concentrations in the amniotic fluid and in the umbilical blood affect pup survival and growth?  Did the cortisol concentrations in the amniotic fluid and pup urine affect pup survival and growth? You can test for multicollinearity. You can use the multicomp R package’s ghlt function to assess post-analysis comparisons. The lme4 R package by Ben Bolker is quite useful for generalized linear mixed model, and the lmerTest R package is useful to calculate the p values.  You have to test for the assumptions.  The QCBS R workshops are very useful resources https://wiki.qcbs.ca/r for learning to run such models with examples data sets and R scripts.  You could report

For each specific concentration, i.e. lactate, glucose and cortisol, you could run generalized linear mixed models (3 models, 1 per molecule) with fixed (e.g. parturition type, categorical variable umbilical blood or amniotic fluid (for lactate and glucose), categorical variable for amniotic fluid or urine (for cortisol), litter size, birth mass and sex of pups, maternal mass, age and parturition experience and random effects (pups and mothers). 

For each of these models, you can i) test for multicollinearity, ii) use the multicomp R package’s ghlt function to assess post-analysis comparisons, and iii) generated the predicted values of the response variables with the predict function or with the effects R package’s.

These are examples of models you could run: you build and run your own models as you see fit with your response variables of interest.

Results

8) My major comment about the statistical models (see above) would affect the results and the discussion. 

9) Results are missing the test statistics, and standard error when applicable.  For the suggested statistical models, you should report the estimates/coefficients ± SE, z or t statistics, and p values.  For a semi-parametric Cox proportional model, you can report the hazard ratio with 95% confidence interval for each fixed effect.  You can also report the survival and hazard functions over time for different variables, which are more informative than a bar graph.

10) Report the biological significance of the results.  You can report the estimated differences. The results only report p values.  Statistical and biological significance can differ.  In the methods, significance is described as p < 0.05, but there is no mention of biological significance in the methods and in the results.

Discussion

11) There is a strong possibility that the discussion would change a lot, if my suggested analyses were run.

12) Lines 530-532: “The correlation of cortisol concentrations in amniotic fluid and fetal urine is very strong, which is consistent with the results of a previous study, where concentrations in amniotic fluid and allantois were also strongly positively correlated.” See my first major comment.

13) Lines 445-456: “The reason for the lower Apgar scores could be influenced by the determination of the parturition date, although we used several parameters to determine the best timing for EL-CS. However, their accuracy depends on several factors such as breed, gestational age and litter size. Therefore, it is challenging to choose the most appropriate parameter for all breeds and each individual animal. Currently, the most accurate method of predicting parturition date is the prepartum progesterone drop, but the use of ultrasound parameters throughout gestation is still necessary to detect bitches before the onset of parturition [33]. An additional factor was the time that elapsed between neonatal extraction and the start of supportive measures, which was minimally prolonged due to obtaining samples for the study. Although this was minimal, it may have contributed to Apgar scores in highly sensitive, slightly immature EL-CS neonates.” It appears that the authors have the data to assess this in one generalized linear mixed model.  If so, the authors could discuss their results rather than speculate.

Conclusion

14) The following seems to me that it would be possible to assess with the data used for this manuscript, lines 547-549: “Further studies are needed to determine whether these two parameters can also be used as predictors of neonatal survival during the first weeks of life, when mortality is the highest.” If the data is available, e.g. death date, this should be assessed as part of this manuscript. There is reason to assume that this data may be available, since the authors state that dogs were weighed once per week for 8 weeks (there were follow-ups) on lines 378-379: “The growth rate of the puppies was followed until they were eight weeks old”.

Minor comments

Lines 197-198: “Urine was immediately frozen at 80 °C …” Add “-” for -80°C

The sentences on lines 224-232 are methods, not results. These sentences are descriptive information about the sample, which are part of the methods. Add the descriptive statistics about the mothers, e.g. mean mass pre-parturition (with standard errors), parturition experience, mean litter size ± standard errors.  How many pups in the study have the same mother and were born in the same litter?  Were mothers repeatedly sampled between litters? Add the sample size of pups, the number of mothers, information of the breeds, and how many pups shared the same mother from the same and different parturitions in the methods.  

The range and distribution of the dates of birth are missing.  For example, this could affect seasonal differences in faecal cortisol metabolite concentrations (see my first major comment).

I hope that my review help to improve the manuscript.  There is a lot in the manuscript that can have a good effect on pup neonatal assessment, pup survival and pup growth.

Sincerely,

Sacha C. Engelhardt, postdoc at the University of Bern

Reviewer 3 Report

This is a very interesting and important study, and the authors are presenting a lot of useful data that improves our knowledge in canine neonatology. However, the presentation should be improved to make the manuscript more concise and easier to read. I also have other questions, comments and suggestions that I would like to authors to consider and address.

Major points of criticism:

The authors were, I assume, sampling several puppies from the same bitch. If this is true, the effect of the bitch should be included in the statistical analysis, otherwise the results are biased. Furthermore, we know that maternal serum glucose levels affect fetal amniotic fluid glucose concentrations, and that maternal size affects serum glucose levels in parturient bitches (Balogh et al., Anim Reprod Sci. 2018;193:209-216.). This should be taken into account, as it may influence glucose concentrations measured in fetal amniotic fluid and umbilical blood, and not only the type of parturition. Have the authors tried to include bitch size in the statistical analysis?

The authors do not give details on the timing and decisions for the elective C-sections, but they discuss immaturity in EL-CS neonates (line 455). Was the date of ovulation known for the bitches? Describe how the date of ovulation was determined from serum P4 levels. On which day after ovulation were the elective C-sections performed? Any other ultrasonographic diagnostics performed to assess fetal maturity prior to C-section? Were any of the puppies premature in this group? Please clarify and add.

In line 145-148 the authors describe that if there was no mammary secretion, puppies were fed with milk replacer every 2 hours. Were these bitches and puppies included in the study? Not having colostrum and milk predisposes to neonatal morbidity and mortality independent of the biomarkers the authors examined or the Apgar scores, and feeding a milk replacer will artificially influence puppy growth rate, all these introducing bias to the authors’ results.

Minor points:

The English language of the manuscript needs minor improvement.

Line 66-83: In the Introduction section the authors should focus and review the literature in more details on what we already know about these prognostic parameters, e.g. glucose, cortisol, lactate on canine neonatal viability and survival, to put the authors' study into perspective. There are several other papers that, in my opinion, should be included for a more comprehensive review of this literature. For example:

Groppetti et al., Theriogenology. 2010;74(7):1187-96.; Mila et al., Prev Vet Med. 2017;143:11-20.; Castagnetti et al., Theriogenology. 2017;94:100-104.

Line 93: Please give approval/permission number and the accurate name of the evaluation committee.

Line 95: Please explain what do you mean by FDA approved food. What about AAFCO approved diet?

Line 97-98: Please add the number of bitches included in each group, i.e. vaginal, elective and emergency CS. The authors present this information in the results “Basic information”, but these data should go up to here to Mat&Meth. Regarding Table 3, I do not understand how the numbers and % were derived. The number of bitches in each breed size group, their age, parity, and litter size/or no. of male and female puppies should be included. It would be a more informative Table 3.

Based on what criteria were bitches selected for elective C-section and how was the timing of the elective C-sections performed (see comments above)? Also, please give more details for the bitches with emergency C-sections regarding their general condition, causes of the C-section, and fetal viability before the C-section.

Line 99-108: This information should be moved to the section on the puppies and Apgar scoring.

Line 111: Describe these criteria for EM-CS

Line 119-124: It is not needed to describe the disinfection protocol so detailed.

Line 135-138: Please describe how were the amniotic fluid samples taken.

Please clarify from which puppies were fetal fluids collected in a bitch, i.e. all puppies? Also, were samples collected from live and stillborn puppies as well? How did the authors identify which amniotic fluid belonged to which puppy? How fast after extraction of the puppies was the umbilical blood collected? Add this information, as the timing may influence lactate concentrations. Also, when, in relation to Apgar scoring was the umbilical blood collected, e.g., before Apgar 5 or 15, etc, and was it at the same time in relation to Apgar scoring for all puppies? Please add this information.

Line 142: Instead of IE, IU = international unit

Line 144-150: It is not needed to give that much detail; the abdomen was routinely closed in three layers. Local anesthetic (lidocain at 1mg/kg) was used...etc.

Line 169: “stimulus on the back” - Why didn't the authors use the toe pinch?

Table 2 is redundant, delete. The authors adequately described the scoring in the text.

Line 189-190: How was specific gravity measured? Include. What was the measuring range for glucose and lactate, and the CVs? What was the sensitivity, measuring range, CVs for cortisol? Did the authors validate the cortisol measurement for the amniotic fluid and for the urine?

Line 206: Birth weight instead of “parturition weight”

Line 211-221: What is “basic descriptive statistics”, please specify. What do the authors mean by vitality and viability? I suggest using only one of these formulations throughout the whole manuscript including figures and tables to be consistent.

Table 4: Proportion of puppies is not needed and should be deleted. I recommend changing the table in that the number of puppies, their gender and survival is presented in relation to the type of parturition. This would be more informative.

Table 5: You can write only the minutes, i.e. 5 min., etc in the first column under Apgar score instead of Apgar 5, etc.. In the first line, I recommend changing "Puppies showing ... distress to just "Severe distress (0-3), %" and so on, and I would add the Apgar score range to each distress category. The last column on Total puppy numbers should be deleted.

Table 6: To be clear, please add in the title what was evaluated as response, i.e. nursing, rooting, righting reflexes

Table 7: this table should be deleted. All this information is available in more details in Table 6 already. The manuscript is long enough with figures and tables, and should be shortened to be more concise.

There are already a lot of figures in the manuscript. I suggest presenting data from Figure 1 and 2 in one combined figure. Also, Figure 7 and 8, amniotic fluid cortisol and urine cortisol, could be presented in one figure, especially that they are depicting the same trend. Figure 9 and 10, amniotic fluid lactate and glucose should also be presented in 1 figure. In all figure legends where relevant, the authors should describe what the box plots, lines and whiskers denote. The number of puppies in each parturition group should be also included in the figure legend.

There is redundancy between Figures 1,2, 3 and 4, and the content of Figures 3-4 is already described in the text. I suggest the authors keep Figure 1 and 2 and combine them into one figure, delete figures 3 and 4, and keep the description in the text (Lines 278-286). I would describe in this text the percentage of severely, moderately, and not distressed pups in EL-CS. You can also add here the % of no distress pups at Apgar60 in each parturition group.

Figure 5 and 6 are misleading, because the number of puppies per column (alive, died) is not given, and it looks like it was the same number in each group. Instead of figure 5 and 6, this information can be given in a table format with both puppy numbers and %.The authors should also present these data for the 60 min. Apgar evaluation for completeness and to see how 60 min. Apgar relates to puppy survival.

Line 273-277: The results of the Fisher's test do not add new information to the data that is presented from the group comparison statistics; The Fisher test is not needed, this information should be deleted.

Line 298-304: Delete this text and just refer to what data can be found in Table 6. Table 7 should be deleted.

Line 312-315: How do your results on specific gravity compare to other studies on canine amniotic fluid? 

Line 365-381: Results on puppy growth rate should be presented after the correlations table in 3.5

Line 366-368: Are the authors presenting results in relation to birth weight? Are litters where all or some puppies were supplemented with milk replacer excluded?  Also, presenting data with only distinguishing between parturition type and not taking the size of the puppies into account may be biased. Were puppy growth rates similar or different between breed size of the dams?

Line 369-375: Please Reformulate this section with regards to above comments and also, if a finding is not statistically significant, then the authors should acknowledge that there is no difference.

Correlation analysis, Line 385-400: Why do the authors use only 5 and 15 min Apgar scores and not Apgar 60? I recommend doing the correlation analysis also for Apgar60. Please refrain from using highly statistically significant; it is either significant or not.

Table 8: I suggest using shorter words in the table, e.g. AM glucose, AM lactate, urine cortisol, UB lactate, UB glucose, etc., and explain abbreviations in the footnote e.g. AM= amniotic fluid, UB=umbilical blood. The authors may want to use abbreviations e.g. AM or UB, throughout the manuscript text.  What is relative growth rate? Is that puppy growth rate and how is that calculated-please clarify.

Line 475: Which increased number of tests?

Line 490-491: This is new information presented here but not in the results. The authors may want to reference this in the Results to be able to discuss it here.

Line 503-504: Hypoglycemia is not that common in parturient/dystocic bitches, rather they can be normo- or hyperglycemic due to stress of parturition, and only a small proportion actually presents with hypoglycemia. See for example these studies:

Frehner et al., Reprod Domest Anim. 2018;53(3):680-687.; Bergstrom et al., Journal of Small Animal Practice, 2006;47:456–460.; Lucio et al., Reprod Domest Anim, 2009;44(Suppl 2), 133–136.; Balogh et al., Anim Reprod Sci. 2018;193:209-216.

Line 508-509: Were the bitches in your study fasted? If yes, for how long? Please also add that to Materials and Methods.

Line 511: I have not seen that the authors reported correlations between mortality and glucose levels.

Line 512-514: The authors should explain better what their results mean in relation to Apgar score, amniotic/umbilical/urinary biomarkers and neonatal survival based on their results to emphasize the importance of their data and their relevance to clinical practice and science. Also, do we know what the normal value for these biomarkers are for healthy, normal canine neonates? Can we say they are hypoglycemic based on adult reference ranges? We know fetuses/neonates can use lactate or BHB or NEFA and not only glucose. Also, please discuss how your results compare to other studies in the veterinary perinatology literature on these biomarkers in these samples?

Conclusion: The conclusion is weak. What is the major message and conclusion of the authors' study? How do these, or can these biomarkers the authors studied complement, or potentially replace Apgar scoring and neonate body weight monitoring to identify and manage at-risk puppies?